# OATS: Outlier-Aware Pruning through Sparse and Low Rank Decomposition

**Stephen Zhang**
University of Toronto
`stephenn.zhang@mail.utoronto.ca`

**Vardan Papyan**
University of Toronto
`vardan.papyan@utoronto.ca`

## Abstract

The recent paradigm shift to large-scale foundation models has brought about a new era for deep learning that, while has found great success in practice, has also been plagued by prohibitively expensive costs in terms of high memory consumption and compute. To mitigate these issues, there has been a concerted effort in post-hoc neural network pruning techniques that do not require costly retraining. Despite the considerable progress being made, existing methods often exhibit a steady drop in model performance as the compression increases. In this paper, we present a novel approach to compressing large transformers, coined OATS, that compresses the model weights by approximating each weight matrix as the sum of a sparse matrix and a low-rank matrix. Prior to the decomposition, the weights are first scaled by the second moment of their input embeddings, so as to ensure the preservation of outlier features recently observed in large transformer models. Without retraining, OATS achieves state-of-the-art performance when compressing large language models, such as Llama-3 and Phi-3, and vision transformers, such as Google's ViT and DINOv2, by up to $60\%$, all while speeding up the model's inference on a CPU by up to $1.37\times$ compared to prior pruning methods. Our code is available at: https://github.com/stephenqz/OATS.

## 1 Introduction

Large scale transformer-based models have found great success in a number of domains ranging from image classification (Wu et al., 2020), language modeling (Devlin et al., 2019), and question answering (Brown et al., 2020). However, these models contain billions of parameters making them computationally expensive to train and deploy, which has lead to an increased demand for resource-saving techniques like model quantization (Dettmers et al., 2022; Egiazarian et al., 2024), parameter efficient fine-tuning (Hu et al., 2022; Zhao et al., 2024b), and, most relevant to this work, neural network pruning (Frantar & Alistarh, 2023).

Pruning has been a key focus for model compression since the early days of deep neural networks (Mozer & Smolensky, 1988; LeCun et al., 1989; Hassibi & Stork, 1992). Various pruning techniques have since emerged, introducing sparsity in the model parameters either before training (Lee et al., 2019; Wang et al., 2020; Tanaka et al., 2020; de Jorge et al., 2021), during training (Zhu & Gupta, 2018; Evci et al., 2020), or post-training (Benbaki et al., 2023). In the context of large foundation models, post-training pruning methods, particularly those requiring minimal (Xia et al., 2022; Ma et al., 2023) or no re-training (Frantar & Alistarh, 2023; Sun et al., 2024b; Ashkboos et al., 2024; Zhang et al., 2024b) are preferred for their computational efficiency. These techniques, when compressing models by $50\%$, have demonstrated the ability to accelerate end-to-end CPU inference by up to $1.8\times$ (Yin et al., 2024b) and GPU inference by up to $1.63\times$ using structured N:M sparsity (Mishra et al., 2021), highlighting their potential in reducing costs during deployment.

Despite the significant advancements in pruning techniques, it was recently shown that current methods suffer from a consistent degradation in model performance as compression levels increase (Yin et al., 2024a). Moreover, although structured pruning offers greater potential for acceleration compared to unstructured pruning, it often imposes a much steeper trade-off in terms of model accuracy and effectiveness (Chen et al., 2022). These challenges underscore the need for more sophisticated pruning strategies that can achieve better performance as compression increases.

## 1.1 CONTRIBUTIONS

To mitigate these issues, we introduce **O**utlier-**A**ware Pruning **T**hrough **S**parse and Low Rank Decomposition (**OATS**): a novel retraining-free method for compressing large transformers that approximates the model's weight matrices as a sum of a sparse matrix and a low-rank matrix. In order to emphasize the outliers recently observed in large transformer models and preserve model performance (Kovaleva et al., 2021; Dettmers et al., 2022; Darcet et al., 2024; Sun et al., 2024a), OATS first scales the weights by the second moment of their corresponding input embeddings.

We evaluate OATS on recent large language models (LLMs) – Phi-3 (Abdin et al., 2024) and Llama-3 (Dubey et al., 2024) – and vision transformers – Google's ViT (Wu et al., 2020) and DinoV2 (Oquab et al., 2023) – demonstrating that OATS achieves new state-of-the-art performance across a wide range of commonly employed performance metrics. Furthermore, by combining structured pruning with unstructured pruning, OATS accelerates CPU inference across all levels of compression when compared to models that utilize just unstructured pruning.

To gain a deeper understanding of the sparse and low-rank terms found by OATS, we split the compressed vision transformers (Wu et al., 2020) into two separate models, a sparse model and a low-rank model, and visualize their respective attention heat maps utilizing attention rollout (Abnar & Zuidema, 2020). These reveal a complementary relationship between the two models, with each focusing on different key areas of the image, effectively segmenting it into distinct regions.

## 2 THE OATS ALGORITHM

The key observation behind the OATS algorithm is that the weight matrices, $W \in \mathbb{R}^{d_{out} \times d_{in}}$, in a transformer model can be faithfully approximated as a summation of a sparse and low-rank matrix by solving the following optimization problem, commonly known as Robust PCA (Chandrasekaran et al., 2009; 2011; Candès et al., 2011):

$$\min_{S,L \in \mathbb{R}^{d_{out} \times d_{in}}} \|W - S - L\|_F^2 \quad \text{s.t.} \quad \text{Rank}(L) \leq r, \ \|S\|_0 \leq k. \tag{1}$$

### 2.1 ALTERNATING THRESHOLDING

To solve Equation 1, OATS leverages the alternating thresholding algorithms proposed by Zhou & Tao (2011), Netrapalli et al. (2014) and Bertsimas et al. (2024) that iteratively alternates between solving for the low-rank term $L$, through singular-value thresholding, and for the sparse term $S$, through hard-thresholding. Given a matrix $A \in \mathbb{R}^{m \times n}$, singular-value thresholding, also known as truncated SVD, is defined as:

$$\text{TRUNCATEDSVD}(A, r) = U_r \Sigma_r V_r^\top,$$

where $U_r, \Sigma_r, V_r^\top$ correspond to the matrices formed by retaining only the top-$r$ singular vectors and singular values from the full SVD of $A$. Hard-thresholding, which succeeds the singular-value thresholding step, is defined as:

$$\text{HARDTHRESHOLD}(A, k) = M \odot A,$$

where $M \in \mathbb{R}^{m \times n}$ is a binary matrix with $k$ non-zero entries coinciding with the $k$ largest entries in magnitude in $A$. These steps are summarized in Algorithm 1 on the right. To optimize memory usage, the low-rank term $L$ is stored through its two low-rank components: $U_r$ and $\Sigma_r V_r^\top$.

### 2.2 ALTERNATIVE SPARSITY PATTERNS

When performing the hard-threshold step, various restrictions can be enforced on the sparsity pattern of the sparse term of the decomposition for enhanced performance or speed-up. The following are two important cases:

---

**Algorithm 1** ALTERNATINGTHRESHOLDING

1: **Inputs:**
2:     Weight Matrix: $W \in \mathbb{R}^{d_{out} \times d_{in}}$
3:     Iterations: $N$
4:     Rank: $r$
5:     Nonzeros: $k$

6: **Procedure:**
7:     $S = 0$
8:     **for** $t = 1$ to $N$ **do**
9:         $L = \text{TRUNCATEDSVD}(W - S, r)$
10:        $S = \text{HARDTHRESHOLD}(W - L, k)$
11:     **end for**
12:     **return**: $S, L$

---

**Row-Wise Thresholding** The hard-thresholding can be performed row-wise rather than layer-wise in which case $M$ would be a binary matrix with $m \cdot \lfloor \frac{k}{m} \rfloor$ non-zero entries coinciding with the $\lfloor \frac{k}{m} \rfloor$ largest entries in magnitude in each row of $A$. Sun et al. (2024b) have shown this leads to better performance.

**N:M Sparsity** The hard-thresholding can be applied at an even more granular level using N:M sparsity, where only the $N$ largest entries by magnitude in every group of $M$ entries in matrix $A$ are nonzero. Recently, NVIDIA's sparse tensor cores have been able to exploit such sparsity patterns for acceleration (Mishra et al., 2021).

## 2.3 INCORPORATING OUTLIER INFORMATION

The alternating thresholding on its own yields suboptimal results because the activations of large-scale transformers exhibit a small number of large-magnitude features and altering these (for example, through the sparse and low-rank approximation) negatively impacts model performance (Kovaleva et al., 2021; Dettmers et al., 2022; Darcet et al., 2024; Sun et al., 2024a). OATS takes inspiration from Wanda (Sun et al., 2024b) and computes a diagonal scaling matrix $D \in \mathbb{R}^{d_{in} \times d_{in}}$ that captures the second moment of the input activations

$$D = \sqrt{\text{diag}(X^\top X)},$$

where $X \in \mathbb{R}^{B \times d_{in}}$ and $B$ is the product of the batch size and sequence length. This diagonal matrix, containing large magnitudes for the outlier features, is used to amplify their significance in the reconstruction error of Equation 1, leading to the following alternative optimization problem:

$$\min_{S, L \in \mathbb{R}^{d_{out} \times d_{in}}} \|WD - S - L\|_F^2 \quad \text{s.t.} \quad \text{Rank}(L) \leq r, \ \|S\|_0 \leq k.$$

The solution of the problem is given by:

$$S, L = \text{ALTERNATINGTHRESHOLDING}\,(WD, N, r, k)$$

which gives a sparse plus low-rank approximation of $WD \approx S + L$. OATS then applies the inverse transformation to reach the final compressed weight:

$$W_{\text{compressed}} = (S + L)D^{-1},$$

where it leverages the fact that $D$ is diagonal so that it both preserves the sparsity pattern of $S$ and is easy to invert. The original weight matrix is replaced with three matrices: the sparse matrix $SD^{-1}$, and two matrices coinciding with the low-rank factorization of $LD^{-1}$. Aligned with Frantar & Alistarh (2023); Sun et al. (2024b), and Zhang et al. (2024b), the activations are calculated through a calibration set that is propagated through the compressed layers.

## 2.4 OATS PARAMETERS

To determine the rank $r$ and the number of nonzeros $k$, OATS takes in as input two hyperparameters: the *compression rate*, $\rho \in (0, 1)$, and the *rank ratio*, $\kappa \in (0, 1)$. The compression rate coincides with the sparsity rate required by existing pruning algorithms and is defined as:

$$\rho = 1 - \frac{\text{\# of nonzero parameters in compressed layer}}{\text{\# of parameters in original layer}} = 1 - \frac{k + r(d_{out} + d_{in})}{d_{out} \cdot d_{in}}.$$

The rank ratio represents the proportion of nonzero parameters that appear in the low-rank term:

$$\kappa = \frac{\text{\# of parameters in low-rank term}}{\text{\# of nonzero parameters in compressed layer}} = \frac{r(d_{out} + d_{in})}{(1 - \rho)d_{out} \cdot d_{in}}.$$

Given a fixed compression rate $\rho$ and rank ratio $\kappa$, the two equations above can be solved to obtain the rank $r$ and nonzeros $k$:

$$r = \left\lfloor \kappa \cdot (1 - \rho) \cdot \frac{d_{out} \cdot d_{in}}{d_{out} + d_{in}} \right\rfloor \qquad k = \lfloor (1 - \kappa) \cdot (1 - \rho) \cdot d_{out} \cdot d_{in} \rfloor. \tag{2}$$

The complete OATS algorithm pseudocode can be found in Algorithm 2 below.

---

**Algorithm 2** OATS

---

1: **Inputs:**
2:     Layer Inputs Propagated Through Prior Compressed Layers: $\boldsymbol{X}^\ell \in \mathbb{R}^{B \times d_{in}}$
3:     Layer Matrix: $\boldsymbol{W}^\ell \in \mathbb{R}^{d_{out} \times d_{in}}$
4:     Compression Rate: $\rho$
5:     Rank Ratio: $\kappa$
6:     Iterations: $N$

7: **Procedure:**
8:     $r \leftarrow \left\lfloor \kappa \cdot (1 - \rho) \cdot \frac{d_{out} \cdot d_{in}}{d_{out} + d_{in}} \right\rfloor , \; k \leftarrow \lfloor (1 - \kappa) \cdot (1 - \rho) \cdot d_{out} \cdot d_{in} \rfloor$
9:     $\boldsymbol{D} \leftarrow \sqrt{\mathrm{diag}(\boldsymbol{X}^\top \boldsymbol{X})}$
10:     $\boldsymbol{L}, \boldsymbol{S} \leftarrow \text{ALTERNATINGTHRESHOLDING}(\boldsymbol{W}\boldsymbol{D}, N, r, k)$
11:     $\boldsymbol{W} \leftarrow (\boldsymbol{L} + \boldsymbol{S})\boldsymbol{D}^{-1}$
12:     **return:** $\boldsymbol{X}^{\ell+1} \leftarrow \boldsymbol{X}^\ell \boldsymbol{W}^\top$

---

## 3 EXPERIMENTS ON LARGE LANGUAGE MODELS

### 3.1 EXPERIMENT SETUP

**Models and Tasks** We evaluate OATS on two state-of-the-art families of LLMs: Phi-3 (Abdin et al., 2024) and Llama-3 (Dubey et al., 2024). To gauge the algorithm's performance under various model sizes, we select Phi-3 Mini, a 3.8B parameter model, Phi-3 Medium, a 14B parameter model, Llama-3 8B, an 8B parameter model, and Llama-3 70B, a 70B parameter model. We utilize LM Harness developed by Gao et al. (2024) to evaluate five-shot performance on the Massive Multitask Language Understanding benchmark by Hendrycks et al. (2021), zero-shot performance on eight tasks, and language generation on WikiText-2.

**Pruning Benchmarks** As OATS does not require costly retraining after model compression, we opt to benchmark it with three current state-of-the-art algorithms that similarly do not require such overhead: SparseGPT by Frantar & Alistarh (2023), Wanda by Sun et al. (2024b), and DSNoT[1] by Zhang et al. (2024b). The parameters utilized for OATS are depicted in Table 1.

| Parameters | Phi-3 | Llama-3 |
|---|---|---|
| Iterations | 80 | 80 |
| Rank Ratio | 25% | 30% |

Table 1: Hyperparameters utilized for OATS across model families. Both parameters are further ablated in Section 3.3.

**Calibration Data** Remaining consistent with Frantar & Alistarh (2023), Sun et al. (2024b), and Zhang et al. (2024b), our calibration data consists of 128 sequences of length 2048 sampled from the first shard of the C4 training set (Raffel et al., 2020). To ensure consistency, we utilize the same calibration data for all pruning algorithms that we benchmark.

**Layer-Wise Compression Rates** We benchmark our algorithm across a wide range of compression rates: $\{0.3, 0.4, 0.5, 0.6\}$. For compression rates at or below $0.5$, we compress all transformer blocks uniformly. At the higher compression rate of $0.6$, we utilize Outlier Weighed Layerwise Sparsity Ratios (OWL) proposed by Yin et al. (2024b) which were shown to lead to significant performance improvements at higher compression rates. All linear layers in a transformer block are pruned uniformly to achieve the desired sparsity rate. We exclude pruning any linear layers that are present in the model head and embeddings which conforms with prior works by Frantar & Alistarh (2023), Sun et al. (2024b), and Zhang et al. (2024b).

**Hardware Speedup** We benchmark the CPU speedup of OATS over its competitors using the DeepSparse Inference Engine developed by NeuralMagic (2021). For GPU speed-up, we include structured N:M sparsity experiments where the rank ratio is varied to measure the trade-off between compression and performance.

---

[1]DSNoT experiments are run with both SparseGPT and Wanda. We report the best results across the two. Further details are in Appendix A.14.

## 3.2 RESULTS

**Five-shot MMLU**  Table 2, below, reports the MMLU accuracy of OATS relative to current state-of-the-art pruning algorithms. OATS is able to outperform all prior methods, across all compression rates, with an increasing gap as the compression rate increases. Notably, at $50\%$ compression, OATS surpasses previous pruning algorithms by a margin of $5.42\%$ on Phi-3 Mini, $2.52\%$ on Phi-3 Medium, $2.86\%$ on Llama-3 8B, and $2.03\%$ on Llama-3 70B.

| Compression | Method | Phi-3 | | Llama-3 | |
|---|---|---|---|---|---|
| | | Mini (3.8B) | Medium (14B) | 8B | 70B |
| 0% | Dense | 70.34 | 76.78 | 64.97 | 79.63 |
| 30% | SparseGPT | 68.31 | 74.12 | 64.25 | 78.28 |
| | Wanda | 67.63 | 75.18 | 63.67 | **79.15** |
| | DSNoT | 68.02 | 75.13 | 63.72 | 79.00 |
| | OATS | **68.84** | **76.15** | **65.22** | 78.47 |
| 40% | SparseGPT | 63.47 | 72.42 | 60.91 | 76.29 |
| | Wanda | 64.15 | 73.34 | 60.33 | 77.16 |
| | DSNoT | 63.57 | 73.20 | 59.99 | 77.70 |
| | OATS | **65.75** | **74.99** | **62.46** | **77.89** |
| 50% | SparseGPT | 53.22 | 67.63 | 53.60 | 72.47 |
| | Wanda | 54.57 | 69.76 | 49.83 | 72.04 |
| | DSNoT | 54.28 | 68.65 | 49.20 | 72.76 |
| | OATS | **59.99** | **72.28** | **56.46** | **74.79** |

Table 2: Comparison of average five-shot accuracies (%) on MMLU under different compression rates.

**Zero-shot Tasks**  Table 3, below, reports the zero-shot accuracy of OATS relative to current state-of-the-art pruning algorithms averaged across the following eight commonly used tasks: PIQA (Bisk et al., 2020); HellaSwag (Zellers et al., 2019); Winogrande (Sakaguchi et al., 2021); OpenBookQA (Mihaylov et al., 2018); RTE (Wang et al., 2018); BoolQ (Clark et al., 2019); ARC-e and ARC-c (Clark et al., 2018). Mirroring the trend observed in the five-shot results, the improvement of OATS over prior pruning algorithms increases with compression, culminating in a $2.05\%$ advantage over prior methods when compressing Phi-3 Mini to $50\%$ of its size.

| Compression | Method | Phi-3 | | Llama-3 | |
|---|---|---|---|---|---|
| | | Mini (3.8B) | Medium (14B) | 8B | 70B |
| 0% | Dense | 71.99 | 74.27 | 69.79 | 75.27 |
| 30% | SparseGPT | 70.63 | **74.53** | 69.08 | 75.07 |
| | Wanda | 70.66 | 74.05 | 68.63 | 75.19 |
| | DSNoT | 71.20 | 74.03 | 68.98 | **75.54** |
| | OATS | **71.48** | 74.04 | **69.34** | 75.24 |
| 40% | SparseGPT | 69.18 | 74.40 | 67.58 | 74.63 |
| | Wanda | 68.80 | 73.01 | 67.04 | 74.10 |
| | DSNoT | 69.08 | 72.90 | 66.65 | 74.29 |
| | OATS | **70.04** | **74.46** | **68.68** | **74.88** |
| 50% | SparseGPT | 66.36 | 73.25 | 64.66 | 73.17 |
| | Wanda | 65.03 | 70.96 | 63.27 | 72.85 |
| | DSNoT | 65.33 | 71.12 | 62.74 | 72.91 |
| | OATS | **68.41** | **73.39** | **65.71** | **73.30** |

Table 3: Comparison of average zero-shot accuracies (%) under different compression rates. Task-specific scores can be found in Appendix A.13.

**Generation Task** Table 4, below, reports the WikiText-2 perplexity of OATS relative to current state-of-the-art pruning algorithms. At $50\%$ compression, OATS results in an $8.49\%$ reduction in perplexity on the larger Phi-3 Medium model, and an even larger $8.99\%$, $9.04\%$, and $9.30\%$ reduction on Phi-3 Mini, Llama-3 8B, and Llama-3 70B respectively.

| Compression | Method | Phi-3 | | Llama-3 | |
|---|---|---|---|---|---|
| | | Mini (3.8B) | Medium (14B) | 8B | 70B |
| 0% | Dense | 9.50 | 6.21 | 10.17 | 2.68 |
| 30% | SparseGPT | 11.19 | 7.48 | 9.71 | 3.24 |
| | Wanda | 10.71 | 7.28 | 9.39 | 3.28 |
| | DSNoT | 10.51 | 7.11 | **9.36** | 3.27 |
| | OATS | **10.27** | **6.85** | 9.59 | **3.07** |
| 40% | SparseGPT | 13.03 | 8.52 | 10.01 | 3.99 |
| | Wanda | 12.59 | 8.49 | 9.74 | 4.08 |
| | DSNoT | 12.17 | 8.24 | 9.60 | 4.10 |
| | OATS | **11.53** | **7.70** | **9.24** | **3.68** |
| 50% | SparseGPT | 16.80 | 9.89 | 11.95 | 5.27 |
| | Wanda | 17.23 | 10.12 | 12.36 | 5.38 |
| | DSNoT | 16.68 | 9.96 | 12.41 | 5.58 |
| | OATS | **15.18** | **9.05** | **10.87** | **4.78** |

Table 4: Comparison of perplexity (lower is better) on WikiText-2 under different compression rates.

**Performance Under High Compression** Table 5, on the right, is the 5-shot MMLU accuracy of models compressed to a higher compression rate of $60\%$, utilizing OWL ratios (Yin et al., 2024b). Following the trend above, OATS continues to outperforms prior methods by a margin of $6.39\%$ on Phi-3 Mini, $5.61\%$ on Phi-3 Medium, and $4.98\%$ on Llama-3 8B.

| Method | Phi-3 | | Llama-3 8B |
|---|---|---|---|
| | Mini | Medium | |
| SparseGPT | 46.20 | 57.91 | 39.48 |
| Wanda | 44.22 | 58.49 | 31.20 |
| DSNoT | 44.75 | 58.20 | 33.28 |
| OATS | **52.59** | **64.10** | **44.46** |

Table 5: MMLU accuracy ($\%$) of models compressed by $60\%$ using OWL ratios.

## 3.3 STUDIES AND HYPERPARAMETER EXPLORATION

We conduct ablation studies for OATS, on Phi-3 Mini at $40\%$ compression rate with a rank ratio of $20\%$, to quantify the impact of the following design choices:

- Scaling the weights by the second moment of the input activations, $\boldsymbol{D}$, versus not scaling.

- Pruning the weights per each output row in the matrix versus pruning layer-wise.

The results are shown in Table 6 below:

| Ablation | | MMLU ($\uparrow$) | Zero-shot ($\uparrow$) | Perplexity ($\downarrow$) |
|---|---|---|---|---|
| No Scaling | Layer-Wise | 62.46 | 67.58 | 19.21 |
| | Row-Wise | 65.31 | 68.22 | 18.34 |
| Scaling by $\boldsymbol{D}$ | Layer-Wise | 64.44 | 70.52 | 11.68 |
| | Row-Wise | **65.84** | **70.71** | **11.50** |

Table 6: Ablation results of OATS on Phi-3-Mini, at $40\%$ compression rate, with a rank ratio of $20\%$. Scaling the weights by the second moment of the input activations and pruning row-wise significantly improves the performance of OATS.

In addition to the ablations, we perform additional experiments to examine the impact of the rank ratio and the number of iterations on the performance of OATS. Figure 1 below shows the results.

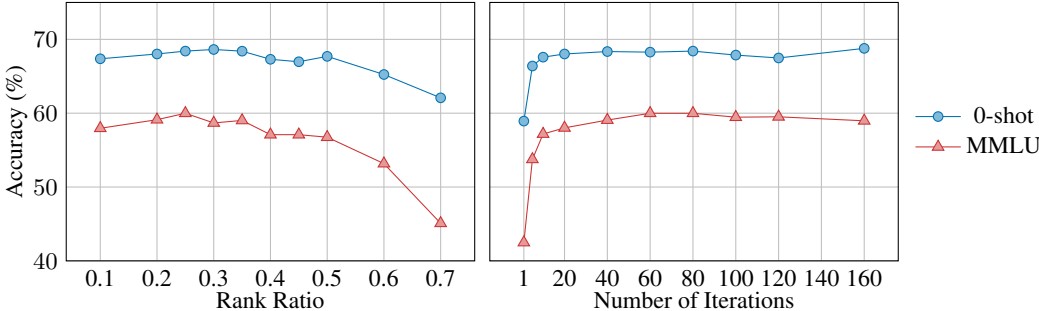

Figure 1: The effect of varying the rank ratio and number of iterations on zero-shot and five-shot accuracy.

The experiments reveal that a rank ratio between $25\%$ to $30\%$ leads to the best performance, with degradation occurring at higher rank ratios. For the number of iterations, performance improves sharply in the first 20 iterations, before leveling off and saturating at around 80 iterations.

## 3.4 HARDWARE SPEEDUP

**CPU Speedup**   We benchmark, using the DeepSparse engine by NeuralMagic (2021), the CPU throughput induced by OATS compared to models pruned with unstructured sparsity. We run end-to-end inference on a compressed Phi-3 Medium 15B model for a single token on an Intel Xeon Gold 6148 CPU @ 2.40GHz with 32 cores (for higher token-counts see Appendix A.6). The achieved throughput and speedup (over a dense model) are shown in Table 7 below. By trading unstructured sparsity for structured sparsity through the low-rank terms, OATS achieves greater CPU speed-up compared to methods that rely solely on unstructured pruning. Notably, at $40\%$ compression, OATS is $1.37\times$ faster than unstructured pruning.

| Compression | Method | Throughput | Speedup |
|---|---|---|---|
| 0% | Dense | 4.03 | $1.00\times$ |
| 30% | Unstructured Pruning | 4.32 | $1.07\times$ |
| | OATS | **5.58** | **$1.38\times$** |
| 40% | Unstructured Pruning | 5.08 | $1.26\times$ |
| | OATS | **6.86** | **$1.73\times$** |
| 50% | Unstructured Pruning | 7.16 | $1.78\times$ |
| | OATS | **8.31** | **$2.06\times$** |

Table 7: Comparison of throughput (tokens/second) and speedup achieved through OATS and unstructured pruning methods relative to their dense counterparts.

**N:M Performance**   We compare the performance of state-of-the-art pruning algorithms, using a 2:4 structured sparsity pattern, with the performance of OATS, using a 2:8 structured sparsity pattern on the sparse term. OATS employs a sparser N:M pattern to compensate for its low-rank term that remains dense. We experiment with rank ratios of $\{0.25, 0.3, 0.35, 0.4, 0.45, 0.5\}$. Unlike previous pruning methods, where N:M structured sparsity enforces a fixed compression rate of $\frac{N}{M}$, OATS allows for a flexible trade-off between compression and model performance by adjusting the rank ratio. Figure 2, below, illustrates the compression ratio against the 5-shot MMLU accuracy for various compression algorithms.

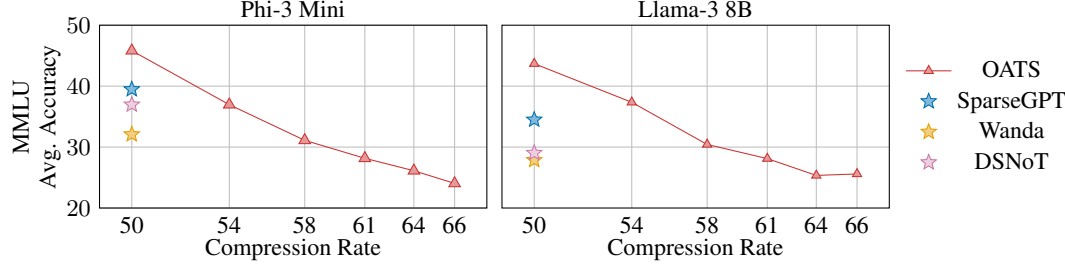

Figure 2: Experiments evaluating OATS with 2:8 structured sparsity on the sparse terms against 2:4 sparsity of state-of-the pruning algorithms. The rank ratio for OATS is varied to capture the performance across different compression rates.

Despite having a sparser structured sparsity pattern of 2:8, OATS is able to recover the model performance through the presence of its low-rank term. Specifically, at a compression rate of $50\%$, OATS is able to outperform all prior state-of-the-art by $6.34\%$ on Phi-3 Mini. In the case of Llama-3 8B, OATS not only surpasses previous methods by $9.2\%$ at $50\%$ compression, but it also outperforms them by $2.86\%$ at an even higher compression rate of $54\%$.

## 4 EXPERIMENTS ON VISION TRANSFORMERS

We run experiments on Google's ViT-Base (Wu et al., 2020), an $86.6$M parameter model trained in a supervised manner on ImageNet-21k (Ridnik et al., 2021) and fine-tuned on ImageNet 2012 (Russakovsky et al., 2015), and DinoV2-Giant (Oquab et al., 2023), a $1.14$B parameter model that was trained through self-supervised learning.

We benchmark OATS against the same three pruning algorithms: SparseGPT, Wanda, and DSNoT, by evaluating top-1 accuracy on the validation set of ImageNet (Russakovsky et al., 2015). A subset of 2048 images from the training set of ImageNet is used for calibration and is maintained consistent across all pruning experiments. All OATS experiments use a rank ratio of $\kappa=20\%$ and $N=80$ iterations. We exclude from compression the embedding and the classifier layers.

The results are shown in Table 8 below. Compared to LLMs, vision transformers show greater resilience to pruning, with DinoV2 experiencing only a $0.41\%$ drop in top-1 accuracy when compressed by $50\%$ using OATS.

| Compression | Method | ViT-Base | DinoV2-Giant |
|---|---|---|---|
| 0% | Dense | 80.33 | 86.55 |
| 30% | SparseGPT | 80.21 | 86.46 |
| | Wanda | **80.28** | 86.47 |
| | DSNoT | 80.16 | 86.46 |
| | OATS | 80.15 | **86.52** |
| 40% | SparseGPT | 79.58 | 86.39 |
| | Wanda | 79.34 | 86.32 |
| | DSNoT | 79.46 | 86.37 |
| | OATS | **79.86** | **86.46** |
| 50% | SparseGPT | 78.44 | 86.04 |
| | Wanda | 76.19 | 85.81 |
| | DSNoT | 76.90 | 85.93 |
| | OATS | **78.77** | **86.14** |

Table 8: ImageNet validation accuracy $(\%)$.

## 5 VISUALIZING AND INTERPRETING THE DECOMPOSITION

To develop a better understanding of how the sparse and low rank components individually contribute to the flow of information through the model, we compute and visualize the attention rollout (Abnar & Zuidema, 2020) of the compressed vision transformers when:

- All low-rank terms are set to zero and inputs are propagated through only the sparse terms.
- All sparse terms are set to zero and inputs are propagated through only the low-rank terms.

Figure 3 below provides a visualization of how the information would flow through a standard transformer block for both settings.

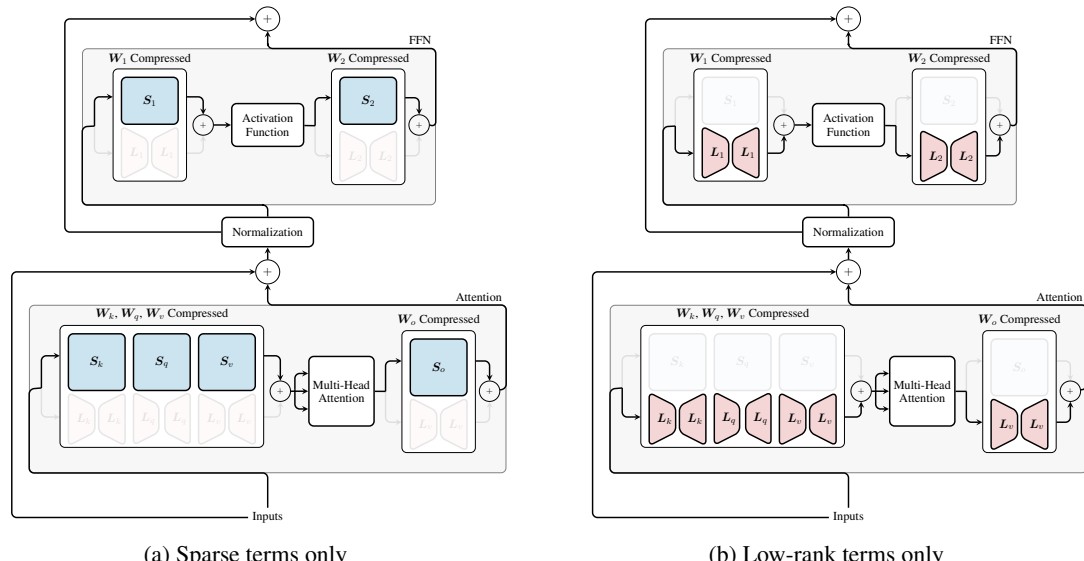

(a) Sparse terms only           (b) Low-rank terms only

Figure 3: A visualization of how the attention rollout is computed to isolate the contribution of the sparse terms versus low-rank terms given by the OATS algorithm.

Figure 4 depicts the attention rollout for various images in the Microsoft COCO dataset (Lin et al., 2014) passed to a ViT-B that was compressed by $50\%$, with a rank ratio of $20\%$.

Original:

Sparse:

Low-Rank:

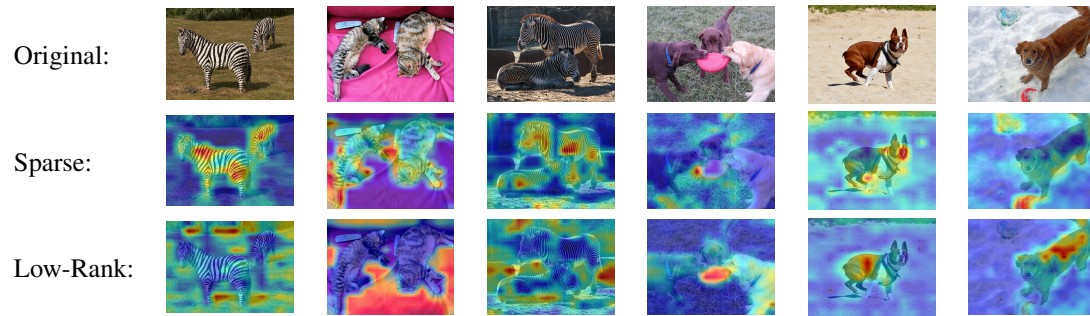

Figure 4: Attention rollout visualization applied to various images on the Microsoft COCO dataset.

The rollout visualizations show that the sparse and low-rank terms capture distinct areas of the image, effectively segmenting it. A careful analysis reveals three distinct partitioning patterns. The first, which is also commonly exhibited in the classical setting (Candès et al., 2011), is when one component (commonly sparse) captures the subject(s), while the other component (commonly low-rank) captures the background. The second is when both components focus on different parts of the

same subject, each capturing distinct features. The third behavior arises when the image contains multiple subjects, with each component isolating a different subject. While these patterns provide initial insights into how the components process visual information, further investigation is needed to fully understand the mechanisms driving these behaviors.

## 6 RELATED WORKS

**Connection with Wanda**   OATS utilizes the same outlier scaling as the the the one that is employed by Wanda (Sun et al., 2024b). In fact, Wanda can be seen as a special case of OATS when the rank ratio $\kappa{=}0$. Indeed, in such a case, according to Equation 2, the low-rank term would become the zero matrix and OATS would perform a single hard thresholding step that is equivalent to the pruning step described by Wanda: $\boldsymbol{W}_{\text{compressed}} = \text{HARDTHRESHOLD}(\boldsymbol{W}\boldsymbol{D}, k)\boldsymbol{D}^{-1}$.

**Sparse and Low-Rank Approximation in Transformers**   The emergence of sparse and low-rank structures in transformers has recently become an area of both theoretical and practical interest. On the theoretical front, Zhao et al. (2024c) showed that the logits of LLMs trained utilizing next token prediction converge to a low rank and sparse structure. On the practical front, Scatterbrain proposed by Chen et al. (2021a) shows that it is possible to approximate the entire attention mechanism with a single sparse and low rank decomposition. Pruning-wise, LoRAP by Li et al. (2024) performs *structured* pruning on the feed-forward linear layers and apply a low-rank decomposition to the attention matrices using a scaling technique similar to OATS.

**Structured Pruning and Low-Rank Adaptation**   Recent works, such as LoSparse (Li et al., 2023), LoRAPrune (Zhang et al., 2024a), and APT (Zhao et al., 2024a), propose variations of applying structured pruning on the weights while incorporating a low-rank adapter that is trained via gradient descent. These are markedly different than OATS, which does not employ any fine-tuning with low-rank adapters, nor does it perform structured pruning (but rather a sparse plus low-rank decomposition which can be thought of as a combination of structured and unstructured pruning).

**Robust PCA Algorithms**   The search for Robust PCA algorithms has been a key area of interest since the inception of the problem. Examples of other approaches include applying a convex relaxation, where the sparsity and low-rank constraints are replaced by $\ell_1$ and nuclear norm surrogates (Zhou et al., 2010), or parameterizing the low-rank matrix as $\boldsymbol{L} = \boldsymbol{U}\boldsymbol{V}^\top$, and applying gradient descent on $\boldsymbol{U}$ and $\boldsymbol{V}$ (Yi et al., 2016; Tong et al., 2021). While OATS utilizes the alternating thresholding approach for its simplicity, future work might want to investigate the use of other algorithms.

**Pruning and Interpretability**   An active area of research is understanding what pruning is pruning and how it impacts model performance. Paganini (2020) show that pruning has a disproportionate negative effect on underrepresented classes. In a similar vein, Yin et al. (2024a) showed that pruning LLMs can irreversibly harm model performance on tasks that are more challenging. We postulate that the low-rank term present in OATS might be able to mitigate the negative impacts of pruning. Indeed, Tables 2 and 5 show that the gap between OATS and prior methods is larger at higher compression, suggesting that the low-rank term plays a critical role in mitigating the loss in performance.

## 7 CONCLUSION

We have introduced OATS, an algorithm that without any re-training, compresses the model's weight matrices through a sparse and low-rank decomposition. Taking inspiration from prior works on the emergence of outlier features, OATS first scales the weights by the second moment of their input embeddings prior to applying an alternating thresholding algorithm. A comprehensive evaluation shows that OATS is able to consistently outperform prior state-of-the-art on various performance metrics across multiple compression rates, models, and modalities, while also improving on CPU speed-up. Beyond just model compression, our visualizations on vision transformers indicate that models exhibit sparse and low-rank structures that capture different segments of the image. This work is the first to reveal the potential of sparse and low-rank decompositions for large-scale transformers, setting the stage for future innovations that can harness this structure to improve model efficiency, performance, and interpretability.

ACKNOWLEDGMENTS

We acknowledge the support of the Natural Sciences and Engineering Research Council of Canada (NSERC). This research was supported in part by the Province of Ontario, the Government of Canada through CIFAR, and industry sponsors of the Vector Institute (`www.vectorinstitute.ai/partnerships/current-partners/`). This research was also enabled in part by support provided by Compute Ontario (`https://www.computeontario.ca`) and the Digital Research Alliance of Canada (`https://alliancecan.ca`).

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

## A APPENDIX

### A.1 ADDITIONAL RELATED WORKS

**Sparse and Low-Rank Decomposition for Pruning** Yu et al. (2017) introduced a method for sparse and low-rank decomposition of CNNs, including AlexNet and GoogLeNet, by solving the

following optimization problem:

$$\min_{\boldsymbol{S},\boldsymbol{L}\in\mathbb{R}^{d_{out}\times d_{in}}} \|\boldsymbol{Y}-(\boldsymbol{S}+\boldsymbol{L})\boldsymbol{X}\|_2^2 \text{ s.t. } \|\boldsymbol{W}-(\boldsymbol{S}+\boldsymbol{L})\|_F^2 \leq \gamma, \text{Rank}(\boldsymbol{L})\leq r, \|\boldsymbol{S}\|_0 \leq k$$

where $\boldsymbol{Y}=\boldsymbol{W}\boldsymbol{X}$. In contrast, OATS employs a different approach, solving:

$$\min_{\boldsymbol{S},\boldsymbol{L}\in\mathbb{R}^{d_{out}\times d_{in}}} \|\boldsymbol{W}-\boldsymbol{S}-\boldsymbol{L}\|_F^2 \text{ s.t. } \text{Rank}(\boldsymbol{L})\leq r, \|\boldsymbol{S}\|_0 \leq k.$$

A key distinction between these methods lies in their objectives: the former directly minimizes reconstruction error, while OATS adopts a simpler formulation. One might question why not follow the approach of minimizing reconstruction error. As noted in DSNoT (Zhang et al., 2024b), pruning methods that prioritize minimizing reconstruction error can degrade model performance in large transformers, particularly in the presence of outlier features. Their findings highlight the importance of avoiding pruning weights within outlier channels. Since feature outliers are a phenomenon unique to large transformer models (Dettmers et al., 2022), this issue would not have been relevant to the work of Yu et al. (2017), which predates the transformer era.

**Pruning Algorithms for Vision Transformers**   There are a number of pruning approaches that have been specifically catered towards pruning vision transformers (Zhu et al., 2021; Chen et al., 2021b; Chavan et al., 2022; Yu et al., 2022a;b; Yu & Xiang, 2023). However, as much of the pruning literature developed on vision transformers involved models of much smaller scale than the large language models employed in this study, almost all of the prominent pruning algorithms require some form of training on the model parameters. As OATS was designed to require no training, OATS and the aforementioned pruning algorithms would not be comparable.

**Low-Rank Adapters during Pre-Training**   In Mozaffari et al. (2024), the authors propose SLOPE, a novel method for accelerating the pre-training phase of LLMs by incorporating N:M sparsity and adding low-rank components to the model weights to enhance model capacity. Similar to OATS, SLOPE leads to a sparse plus low-rank structure in the model's weight matrices, however, the low-rank terms are introduced during the final phase of pre-training and are actively trained on the model loss function. In contrast, OATS is designed as a lightweight method to accelerate inference. OATS does not require any training or fine-tuning, but instead approximates pre-trained weight matrices by solving the Robust PCA problem.

**Quantized Sparse Low-Rank Approximation**   An independent and concurrent work with OATS proposes SLIM (Mozaffari & Dehnavi, 2024), a novel pipeline that combines pruning and quantization. To restore lost performance from compression, SLIM derives a low-rank term using singular-value thresholding and adopts a scaling technique akin to OATS. However, instead of the $L^2$ norm, SLIM utilizes the average absolute value across the batch and sequence dimensions. As a further deviation from OATS, SLIM is also not performing an alternating thresholding algorithm. Instead, they perform a single quantization and pruning step to initialize the quantized and sparse terms, followed by a single singular value thresholding step to establish the low-rank term.

## A.2   TIME COMPLEXITY AND WALL-CLOCK TIME FOR OATS

The time complexity for OATS is $\mathcal{O}(LN\alpha)$ where $L$ is the number of transformer blocks, $N$ is number of iterations, and

$$\alpha = \max_{\boldsymbol{W}} d_{out}^{\boldsymbol{W}} \cdot d_{in}^{\boldsymbol{W}} \cdot r^{\boldsymbol{W}}$$

where the max is taken over the weight matrices, $\boldsymbol{W} \in \mathbb{R}^{d_{out}^{\boldsymbol{W}} \times d_{in}^{\boldsymbol{W}}}$, in a transformer block and $r^{\boldsymbol{W}}$ is the rank of the low-rank term for that weight matrix. The value $\alpha$ represents the time complexity needed to perform the singular value thresholding in OATS.

Table 9 below reports the wall-clock time needed to perform a single iteration of the alternating threshold algorithm for a single transformer block for the different models that were compressed. All experiments utilized a single NVIDIA A40 with 48GB of GPU memory.

| Phi-3 | | Llama-3 | |
|---|---|---|---|
| Mini (3.8B) | Medium (14B) | 8B | 70B |
| 8.85 | 26.02 | 17.10 | 152.80 |

Table 9: Wall-clock time (in seconds) needed to perform a single iteration of the alternating projection algorithm in OATS.

While OATS does require more wall-clock time than prior pruning algorithms, in practice, model compression would only need to be performed once before deployment. This trade-off is therefore worthwhile given the substantial performance improvements, particularly on more challenging tasks like MMLU (see Table 2). Furthermore, like prior pruning algorithms, compressing the layers within a single transformer block can be done in parallel. For example, the time needed per transformer block of Llama-3 70B can be reduced to 71.10 seconds by compressing in parallel across four NVIDIA A40 GPUs.

The total wall-clock time can also be reduced by lowering the number of OATS iterations. Presented in Table 10 is an exploratory experiment compressing Llama-3 70B by 50% with a rank ratio of 0.3 with only 20 iterations. Even with only a quarter of the iterations, OATS is still able to outperform all prior pruning algorithms across all performance metrics.

| MMLU ($\uparrow$) | Zero-shot ($\uparrow$) | Perplexity ($\downarrow$) |
|---|---|---|
| 74.02 | 73.41 | 4.95 |

Table 10: Exploratory experiment measuring the performance of OATS on Llama-3 70B with only 20 iterations.

### A.3 USING A ROBUST SCALING MATRIX

To explore whether the scaling matrix $\boldsymbol{D}$ is truly related to the outlier information, we run the following two experiments:

- Scaling by the square root of the features' second moments, as is currently done in OATS.

- Scaling by the median of the features' absolute values (computed along batch and sequence dimensions):
$$\boldsymbol{D}_{robust} = \text{median}(|X|)$$

The second experiment estimates the square root of the second moment of features in a manner that is robust (insensitive) to outliers akin to the Median Absolute Deviation estimator from the robust statistics literature (Huber, 1981). The results of the two experiments are presented in Table 11 below:

| Scaling Matrix | MMLU ($\uparrow$) | Zero-shot ($\uparrow$) | Perplexity ($\downarrow$) |
|---|---|---|---|
| $\boldsymbol{D}_{robust}$ | 55.54 | 65.77 | 18.59 |
| $\boldsymbol{D}$ | **59.99** | **68.41** | **15.18** |

Table 11: Results of OATS on Phi-3-Mini, at $50\%$ compression rate, with a rank ratio of $25\%$ using different scaling matrices.

The findings show that using the robust scaling method results in significantly worse performance. Hence, the scaling matrix $\boldsymbol{D}$ that is sensitive to the outlier features and captures their scale leads to better compression.

## A.4 SWITCHING THE ORDER OF THRESHOLDING

OATS opts to perform the singular-value thresholding first followed by the hard thresholding similar to Zhou & Tao (2011). However, one might consider whether the alternative order could lead to faster convergence or a better approximation. Presented in Table 12 below is an extension of the ablation studies presented in Section 3.3, reporting the performance of OATS where the hard-thresholding is performed first:

| **First Thresholding Operation** | **MMLU** ($\uparrow$) | **Zero-shot** ($\uparrow$) | **Perplexity** ($\downarrow$) |
|---|---|---|---|
| Hard-Thresholding | 65.51 | 70.54 | 11.72 |
| Singular Value Thresholding (OATS) | **65.84** | **70.71** | **11.50** |

Table 12: Ablation results of switching of the order between the two thresholding operations. Experiments were run on Phi-3-Mini, at $40\%$ compression rate, with a rank ratio of $20\%$.

While the performance still remains competitive, across all performance metrics, the switched order falls short of matching the original order presented in Algorithm 1.

## A.5 MAGNITUDE-BASED PRUNING FOR THE SPARSE COMPONENT

Another question that we explored is whether it is sufficient to capture the outlier information entirely in the low-rank term and determine the sparse term through a hard-thresholding that does not depend on the scaling:

$$\boldsymbol{S} = \text{HARDTHRESHOLD}((\boldsymbol{W}\boldsymbol{D} - \boldsymbol{L})\boldsymbol{D}^{-1}, k).$$

Presented in Table 13 below are the results:

| **Outlier Scaling** | **MMLU** ($\uparrow$) | **Zero-shot** ($\uparrow$) | **Perplexity** ($\downarrow$) |
|---|---|---|---|
| Low-Rank Term Only | 65.22 | **71.01** | 12.49 |
| Both Terms (OATS) | **65.84** | 70.71 | **11.50** |

Table 13: Ablation results of OATS on Phi-3-Mini, at $40\%$ compression rate, with a rank ratio of $20\%$ testing whether the outlier information can be entirely captured by the low-rank term.

## A.6 ADDITIONAL CPU SPEEDUP EXPERIMENTS

We provide additional experiments measuring the CPU speedup of OATS compared to dense models. Following the same setup as Section 3.4, we run end-to-end inference on a compressed Phi-3 Medium 15B model, *but instead with a sequence consisting of 256 tokens*, on an Intel Xeon Gold 6148 CPU @ 2.40GHz with 32 cores. The results are presented in Table 14 below:

| Compression | Method | Throughput | Speedup |
|---|---|---|---|
| 0% | Dense | 0.37 | $1.00\times$ |
| 30% | Unstructured Pruning | 0.40 | $1.08\times$ |
| | OATS | 0.40 | $1.08\times$ |
| 40% | Unstructured Pruning | 0.43 | $1.16\times$ |
| | OATS | 0.43 | $1.16\times$ |
| 50% | Unstructured Pruning | 0.49 | $1.32\times$ |
| | OATS | 0.49 | $1.32\times$ |

Table 14: Comparison of throughput (tokens/second) and speedup achieved through OATS and unstructured pruning methods relative to their dense counterparts on a sequence consisting of 256 tokens.

Although OATS and unstructured pruning now deliver comparable CPU speed-ups, we expect further gains are achievable by designing optimizations that explicitly exploit OATS' sparse plus low-rank structure.

## A.7 ADDITIONAL HYPERPARAMETER TESTS FOR OATS

Presented in Table 15 below includes more hyperparameters that we experimented with for the Phi-3 Mini and Llama-3 8B models.

| Model | Compression | Rank Ratio | MMLU (↑) | Zero-Shot (↑) | Perplexity (↓) |
|-------|-------------|------------|----------|---------------|----------------|
| Phi-3 Mini | 30% | 0.1 | 68.70 | 71.65 | 10.24 |
| | | 0.2 | 68.02 | 71.81 | 10.21 |
| | | 0.3 | 69.28 | 72.07 | 10.28 |
| | 40% | 0.1 | 65.75 | 69.94 | 11.57 |
| | | 0.2 | 65.84 | 70.71 | 11.50 |
| | | 0.3 | 66.81 | 70.54 | 11.60 |
| | 50% | 0.1 | 57.96 | 67.37 | 15.48 |
| | | 0.2 | 59.12 | 68.02 | 15.13 |
| | | 0.3 | 58.68 | 68.63 | 15.47 |
| Llama-3 8B | 30% | 0.1 | 63.62 | 68.99 | 9.35 |
| | | 0.2 | 63.09 | 69.54 | 9.09 |
| | 40% | 0.1 | 61.44 | 68.23 | 9.23 |
| | | 0.2 | 61.97 | 68.43 | 9.09 |
| | 50% | 0.1 | 56.46 | 65.33 | 10.85 |
| | | 0.2 | 56.07 | 65.51 | 10.70 |

Table 15: Further experiments testing different hyperparameter configurations for OATS on the Phi-3 Mini and Llama-3 8B models.

## A.8 PERFORMANCE GAP BETWEEN OATS AND WANDA

To better understand the increase in performance induced by the addition of the low-rank term in OATS, we have compiled in Table 16 below the performance gaps between OATS and Wanda.

| Model | Compression | MMLU (↑) | Zero-Shot (↑) | Perplexity (↓) |
|-------|-------------|----------|---------------|----------------|
| Phi-3 Mini | 30% | +1.21% | +0.82% | -0.44 |
| | 40% | +1.60% | +1.24% | -1.06 |
| | 50% | +5.42% | +3.38% | -2.05 |
| Phi-3 Medium | 30% | +0.97% | -0.01% | -0.43 |
| | 40% | +1.65% | +1.45% | -0.79 |
| | 50% | +2.52% | +2.43% | -1.07 |
| Llama-3 8B | 30% | +1.55% | +0.71% | +0.20 |
| | 40% | +2.13% | +1.64% | -0.50 |
| | 50% | +6.63% | +2.44% | -1.49 |
| Llama-3 70B | 30% | -0.68% | +0.05% | -0.17 |
| | 40% | +0.73% | +0.78% | -0.40 |
| | 50% | +2.74% | +0.45% | -0.60 |

Table 16: The impact of including a low-rank term in OATS compared to Wanda.

## A.9 QWEN 2.5 EXPERIMENTS

Presented in Table 17 below are additional experiments benchmarking OATS against prior pruning algorithms on the Qwen 2.5 3B Instruct model (Qwen Team, 2024). All OATS experiments utilize a rank ratio of $0.2$ and $80$ iterations.

| Compression | Method | MMLU (↑) | Zero-Shot (↑) | Perplexity (↓) |
|---|---|---|---|---|
| 0% | Dense | 65.99 | 68.49 | 11.02 |
| 30% | SparseGPT | **65.65** | 67.91 | 11.55 |
|  | Wanda | 65.46 | 68.08 | 11.66 |
|  | DSNoT | 65.65 | 68.21 | 11.67 |
|  | OATS | 65.36 | **68.74** | **11.45** |
| 40% | SparseGPT | 63.04 | 67.64 | 12.56 |
|  | Wanda | 61.88 | 67.14 | 12.89 |
|  | DSNoT | 62.26 | 67.42 | 12.91 |
|  | OATS | **64.30** | **68.76** | **12.31** |
| 50% | SparseGPT | 57.43 | 64.36 | 14.92 |
|  | Wanda | 55.39 | 64.10 | 16.27 |
|  | DSNoT | 55.78 | 64.77 | 16.43 |
|  | OATS | **58.78** | **65.74** | **14.91** |

Table 17: Benchmarks for OATS on the Qwen 2.5 3B Instruct model.

## A.10 MMLU SUBJECTS

We evaluate on the following MMLU subjects:

- Abstract Algebra

- Business Ethics

- College Computer Science

- College Mathematics

- Conceptual Physics

- Formal Logic

- Machine Learning

- Miscellaneous

- Philosophy

- Global Facts

which aligns with the subset utilized in the codebase of Ashkboos et al. (2024) that can be found here: https://github.com/microsoft/TransformerCompression.

## A.11 ATTENTION ROLLOUT: DETAILS

To generate the attention rollout visualizations depicted in Section 5, we average the attention matrices across the attention heads and discard the bottom $40\%$ attention pixels. The act of discarding the lowest value attention pixels was inspired by the following blog post by Gil (2021).

## A.12 ZERO-SHOT TASK-SPECIFIC PERFORMANCE

Table 18, below, shows the task-specific performance for the zero-shot evaluation results presented in Section 3.2 and Appendix A.14.

| Model | Compression | Method | PIQA | HellaSwag | WinoGrande | OpenBookQA | RTE | BoolQ | ARC-e | ARC-c |
|---|---|---|---|---|---|---|---|---|---|---|
| Phi-3 Mini | 0% | Dense | 81.23 | 77.50 | 73.56 | 46.80 | 75.81 | 85.32 | 78.45 | 57.25 |
| | 30% | SparseGPT | 78.94 | 76.94 | 69.85 | 49.60 | 73.29 | 84.13 | 76.39 | 55.89 |
| | | Wanda | 79.65 | 76.27 | 71.59 | 48.00 | 73.65 | 83.70 | 77.23 | 55.20 |
| | | DSNoT w/ SparseGPT | 80.09 | 75.61 | 72.22 | 47.40 | 74.37 | 84.22 | 77.86 | 54.69 |
| | | DSNoT w/ Wanda | 80.41 | 75.52 | 72.06 | 47.60 | 74.37 | 84.53 | 79.55 | 55.55 |
| | | OATS | 80.03 | 77.07 | 72.61 | 47.60 | 74.37 | 84.92 | 77.44 | 57.76 |
| | 40% | SparseGPT | 78.35 | 75.07 | 68.59 | 47.00 | 72.20 | 83.67 | 75.29 | 53.24 |
| | | Wanda | 78.35 | 73.87 | 69.30 | 45.40 | 71.84 | 83.18 | 76.52 | 51.96 |
| | | DSNoT w/ SparseGPT | 78.56 | 72.99 | 70.64 | 46.20 | 70.40 | 82.72 | 76.52 | 52.82 |
| | | DSNoT w/ Wanda | 79.33 | 73.06 | 70.88 | 44.00 | 70.76 | 83.79 | 77.40 | 53.41 |
| | | OATS | 79.38 | 75.86 | 70.01 | 46.60 | 72.56 | 83.98 | 76.85 | 55.12 |
| | 50% | SparseGPT | 77.20 | 70.63 | 66.46 | 45.20 | 70.76 | 83.06 | 70.58 | 47.01 |
| | | Wanda | 76.33 | 67.70 | 66.38 | 41.80 | 66.43 | 81.83 | 72.43 | 47.35 |
| | | DSNoT w/ SparseGPT | 76.28 | 67.16 | 65.90 | 42.20 | 63.90 | 81.56 | 72.90 | 48.04 |
| | | DSNoT w/ Wanda | 75.52 | 66.54 | 67.64 | 43.00 | 65.34 | 82.54 | 73.48 | 48.55 |
| | | OATS | 77.26 | 71.64 | 69.53 | 44.80 | 73.65 | 81.28 | 77.10 | 52.05 |
| Phi-3 Medium | 0% | Dense | 81.66 | 82.83 | 75.85 | 50.00 | 77.62 | 88.17 | 78.41 | 59.64 |
| | 30% | SparseGPT | 81.39 | 82.02 | 75.77 | 50.80 | 77.26 | 87.80 | 80.05 | 61.18 |
| | | Wanda | 81.39 | 80.88 | 76.01 | 49.40 | 76.90 | 87.74 | 79.59 | 60.49 |
| | | DSNoT w/ SparseGPT | 81.94 | 80.76 | 76.95 | 48.40 | 75.81 | 87.65 | 79.25 | 59.81 |
| | | DSNoT w/ Wanda | 81.66 | 81.03 | 77.27 | 49.20 | 76.53 | 87.80 | 78.96 | 59.81 |
| | | OATS | 81.07 | 82.09 | 74.43 | 51.20 | 78.34 | 88.38 | 78.16 | 58.70 |
| | 40% | SparseGPT | 80.41 | 80.70 | 75.53 | 51.20 | 77.26 | 88.32 | 81.23 | 60.58 |
| | | Wanda | 79.87 | 78.15 | 75.45 | 48.60 | 77.26 | 87.71 | 78.11 | 58.96 |
| | | DSNoT w/ SparseGPT | 79.82 | 78.07 | 75.37 | 47.00 | 76.53 | 87.98 | 77.31 | 58.19 |
| | | DSNoT w/ Wanda | 80.30 | 78.11 | 74.66 | 47.80 | 77.26 | 88.04 | 78.11 | 58.87 |
| | | OATS | 81.39 | 81.72 | 75.06 | 51.00 | 77.62 | 87.65 | 80.39 | 60.84 |
| | 50% | SparseGPT | 79.71 | 78.27 | 73.64 | 50.40 | 75.45 | 87.09 | 82.03 | 59.39 |
| | | Wanda | 78.29 | 74.07 | 74.03 | 45.00 | 75.81 | 85.72 | 77.44 | 57.34 |
| | | DSNoT w/ SparseGPT | 79.27 | 74.30 | 74.59 | 44.40 | 76.90 | 85.26 | 77.69 | 56.57 |
| | | DSNoT w/ Wanda | 78.56 | 73.81 | 75.14 | 43.60 | 75.81 | 86.33 | 77.53 | 58.02 |
| | | OATS | 81.07 | 79.18 | 76.09 | 50.20 | 74.73 | 87.77 | 80.05 | 58.02 |
| Llama-3 8B | 0% | Dense | 80.74 | 79.16 | 73.40 | 45.0 0 | 67.87 | 80.98 | 77.69 | 53.50 |
| | 30% | SparseGPT | 80.36 | 78.58 | 73.24 | 44.40 | 66.79 | 81.38 | 76.81 | 51.11 |
| | | Wanda | 79.98 | 78.00 | 73.64 | 44.40 | 64.26 | 81.62 | 76.18 | 50.94 |
| | | DSNoT w/ SparseGPT | 80.20 | 78.12 | 73.80 | 44.40 | 65.70 | 82.20 | 75.72 | 51.71 |
| | | DSNoT w/ Wanda | 79.82 | 77.99 | 73.09 | 44.80 | 63.18 | 81.80 | 77.06 | 51.37 |
| | | OATS | 80.03 | 78.75 | 73.64 | 45.20 | 66.06 | 81.13 | 76.94 | 52.99 |
| | 40% | SparseGPT | 79.16 | 76.74 | 73.32 | 41.80 | 64.26 | 81.31 | 74.71 | 49.32 |
| | | Wanda | 78.73 | 75.90 | 72.22 | 44.40 | 63.18 | 80.46 | 72.31 | 49.15 |
| | | DSNoT w/ SparseGPT | 78.29 | 75.92 | 73.32 | 42.60 | 58.48 | 80.86 | 73.11 | 47.70 |
| | | DSNoT w/ Wanda | 78.51 | 75.52 | 73.24 | 43.80 | 61.73 | 80.70 | 72.01 | 47.70 |
| | | OATS | 79.71 | 77.18 | 74.19 | 43.80 | 67.51 | 82.39 | 74.92 | 49.74 |
| | 50% | SparseGPT | 77.58 | 73.12 | 72.85 | 40.80 | 59.21 | 79.30 | 69.28 | 45.14 |
| | | Wanda | 77.53 | 69.34 | 70.24 | 40.00 | 61.73 | 76.57 | 66.96 | 43.77 |
| | | DSNoT w/ SparseGPT | 76.88 | 69.45 | 69.30 | 39.60 | 59.21 | 77.25 | 67.93 | 43.32 |
| | | DSNoT w/ Wanda | 77.09 | 68.57 | 69.77 | 38.60 | 57.76 | 76.27 | 67.34 | 43.43 |
| | | OATS | 77.75 | 73.17 | 71.74 | 41.00 | 64.98 | 79.66 | 72.35 | 45.05 |
| Llama-3 70B | 0% | Dense | 84.33 | 84.89 | 80.35 | 48.60 | 68.23 | 85.26 | 86.03 | 64.51 |
| | 30% | SparseGPT | 84.66 | 84.63 | 80.35 | 48.00 | 69.31 | 85.26 | 85.02 | 63.31 |
| | | Wanda | 84.39 | 83.97 | 80.58 | 48.40 | 70.04 | 85.29 | 85.06 | 63.82 |
| | | DSNoT w/ SparseGPT | 84.06 | 84.49 | 80.11 | 48.20 | 69.68 | 85.57 | 85.10 | 63.82 |
| | | DSNoT w/ Wanda | 84.55 | 84.48 | 81.22 | 47.80 | 71.12 | 85.93 | 85.06 | 64.16 |
| | | OATS | 84.28 | 84.40 | 80.66 | 48.40 | 69.31 | 85.32 | 85.90 | 63.65 |
| | 40% | SparseGPT | 83.62 | 83.77 | 80.03 | 47.80 | 69.68 | 85.69 | 84.47 | 61.95 |
| | | Wanda | 83.57 | 83.03 | 78.93 | 47.40 | 68.23 | 85.05 | 84.34 | 62.29 |
| | | DSNoT w/ SparseGPT | 82.37 | 83.21 | 78.85 | 46.20 | 66.43 | 85.20 | 83.75 | 60.07 |
| | | DSNoT w/ Wanda | 83.79 | 83.35 | 79.72 | 46.80 | 67.87 | 85.57 | 84.89 | 62.37 |
| | | OATS | 84.44 | 83.69 | 80.11 | 48.60 | 70.40 | 84.56 | 84.55 | 62.71 |
| | 50% | SparseGPT | 83.13 | 81.68 | 79.32 | 46.20 | 71.12 | 85.17 | 81.27 | 57.51 |
| | | Wanda | 83.08 | 81.12 | 78.22 | 48.00 | 69.31 | 84.22 | 81.61 | 57.25 |
| | | DSNoT w/ SparseGPT | 81.34 | 80.68 | 77.82 | 45.60 | 70.04 | 84.62 | 80.98 | 55.12 |
| | | DSNoT w/ Wanda | 85.24 | 81.64 | 78.45 | 46.80 | 69.31 | 85.23 | 81.52 | 57.76 |
| | | OATS | 83.41 | 82.16 | 79.01 | 47.40 | 68.59 | 85.47 | 82.11 | 58.28 |

Table 18: Task-Specific Zero-Shot Results

## A.13 WIKITEXT-2 PERPLEXITY

We utilized LM Harness (Gao et al., 2024) to measure the Wikitext-2 perplexities for the Phi-3 Mini, Medium, and Llama-3 8B models in Table 4. However, prior works (Sun et al., 2024b; Zhang et al., 2024b) have also opted to measure Wikitext-2 perplexities utilizing a different normalization and dataset segmentation. We report the Wikitext-2 perplexities utilizing the latter [2] approach in Table 19 below:

---

[2]We use the implementation sourced from the code base of Sun et al. (2024b): https://github.com/locuslab/wanda/blob/main/lib/eval.py

| Compression | Method | Phi-3 | | Llama-3 |
| --- | --- | --- | --- | --- |
| | | Mini (3.8B) | Medium (14B) | 8B |
| 0% | Dense | 5.64 | 4.02 | 5.54 |
| 30% | SparseGPT | 6.32 | 4.56 | 6.04 |
| | Wanda | 6.19 | 4.46 | 6.03 |
| | DSNoT | 6.11 | 4.40 | 5.99 |
| | OATS | **5.96** | **4.30** | **5.87** |
| 40% | SparseGPT | 7.05 | 5.02 | 6.75 |
| | Wanda | 6.99 | 4.97 | 6.69 |
| | DSNoT | 6.81 | 4.89 | 6.67 |
| | OATS | **6.51** | **4.64** | **6.39** |
| 50% | SparseGPT | 8.55 | 5.65 | 8.30 |
| | Wanda | 8.84 | 5.73 | 8.79 |
| | DSNoT | 8.58 | 5.68 | 8.84 |
| | OATS | **7.98** | **5.28** | **7.77** |

Table 19: Comparison of perplexity (lower is better) utilizing a different normalization and dataset segmentation.

## A.14 PRUNING IMPLEMENTATION AND HYPERPARAMETERS

Our code and pruning method implementations are based on the following codebases:

- SliceGPT (Ashkboos et al., 2024): https://github.com/microsoft/TransformerCompression

- SparseGPT (Frantar & Alistarh, 2023): https://github.com/IST-DASLab/sparsegpt

- Wanda (Sun et al., 2024b): https://github.com/locuslab/wanda

- DSNoT (Zhang et al., 2024b): https://github.com/zyxxmu/DSnoT

- OWL (Yin et al., 2024b): https://github.com/luuyin/OWL

We utilize Huggingface's Transformers library to implement the large language models and vision transformers for our experiments (Wolf et al., 2020).

### A.14.1 SPARSEGPT HYPERPARAMETERS

We utilize a blocksize of 128 across all experiments and a Hessian dampening of $0.01$ and $0.1$ where the ladder is utilized only when faced with non-positive definiteness issues related with the Cholesky decomposition.

### A.14.2 DSNOT HYPERPARAMETERS

We run experiments utilizing DSNoT where the initial masks are generated by SparseGPT and Wanda. All DSNoT experiments were run with $50$ iterations and an update threshold of $0.1$. Table 20, below, shows the results distinguishing between the two initial methods that were utilized.

| Model | Compression | Method | MMLU (↑) | Zero-Shot(↑) | Perplexity(↓) |
|---|---|---|---|---|---|
| Phi-3 Mini | 30% | DSNoT w/ SparseGPT | 67.01 | 70.81 | 10.55 |
| | | DSNoT w/ Wanda | 68.02 | 71.20 | 10.51 |
| | 40% | DSNoT w/ SparseGPT | 62.94 | 68.86 | 12.29 |
| | | DSNoT w/ Wanda | 63.57 | 69.08 | 12.17 |
| | 50% | DSNoT w/ SparseGPT | 53.99 | 64.74 | 16.71 |
| | | DSNoT w/ Wanda | 54.28 | 65.33 | 16.68 |
| Phi-3 Medium | 30% | DSNoT w/ SparseGPT | 74.89 | 73.82 | 7.11 |
| | | DSNoT w/ Wanda | 75.13 | 74.03 | 7.11 |
| | 40% | DSNoT w/ SparseGPT | 73.15 | 72.54 | 8.24 |
| | | DSNoT w/ Wanda | 73.20 | 72.90 | 8.27 |
| | 50% | DSNoT w/ SparseGPT | 68.65 | 71.12 | 9.96 |
| | | DSNoT w/ Wanda | 68.12 | 71.10 | 10.02 |
| Llama-3 8B | 30% | DSNoT w/ SparseGPT | 62.99 | 68.98 | 9.37 |
| | | DSNoT w/ Wanda | 63.72 | 68.64 | 9.36 |
| | 40% | DSNoT w/ SparseGPT | 58.97 | 66.28 | 9.60 |
| | | DSNoT w/ Wanda | 59.99 | 66.65 | 9.68 |
| | 50% | DSNoT w/ SparseGPT | 49.15 | 62.74 | 12.41 |
| | | DSNoT w/ Wanda | 49.20 | 62.35 | 12.42 |
| Llama-3 70B | 30% | DSNoT w/ SparseGPT | 78.76 | 75.13 | 3.28 |
| | | DSNoT w/ Wanda | 79.00 | 75.54 | 3.27 |
| | 40% | DSNoT w/ SparseGPT | 76.39 | 73.26 | 4.16 |
| | | DSNoT w/ Wanda | 77.70 | 74.29 | 4.10 |
| | 50% | DSNoT w/ SparseGPT | 72.18 | 72.02 | 5.87 |
| | | DSNoT w/ Wanda | 72.76 | 72.91 | 5.58 |

Table 20: LLM performance metrics of DSNoT with different initial methods.

Table 21, below, shows the analogous results but for our vision transformer experiments:

| Model | Compression | Method | Accuracy (%) |
|---|---|---|---|
| ViT-Base | 30% | DSNoT w/ SparseGPT | 80.01 |
| | | DSNoT w/ Wanda | 80.16 |
| | 40% | DSNoT w/ SparseGPT | 79.12 |
| | | DSNoT w/ Wanda | 79.46 |
| | 50% | DSNoT w/ SparseGPT | 75.83 |
| | | DSNoT w/ Wanda | 76.90 |
| DinoV2-Giant | 30% | DSNoT w/ SparseGPT | 86.46 |
| | | DSNoT w/ Wanda | 86.45 |
| | 40% | DSNoT w/ SparseGPT | 86.37 |
| | | DSNoT w/ Wanda | 86.30 |
| | 50% | DSNoT w/ SparseGPT | 85.87 |
| | | DSNoT w/ Wanda | 85.93 |

Table 21: ImageNet Validation Accuracy of DSNoT with different initial methods.

