# OpenReview forum: "OATS: Outlier-Aware Pruning Through Sparse and Low Rank Decomposition"
_ICLR.cc/2025/Conference — ICLR 2025 Poster_

### Official Review · Reviewer_rXxS · 2024-10-29

**Soundness:** 3
**Presentation:** 2
**Contribution:** 3
**Rating:** 6
**Confidence:** 4

**Summary:**

In this paper, the author presents a novel approach to compressing large transformers, coined OATS, that utilizes the second moment information in the input embeddings to decompose the model weights into a sum of sparse and low-rank matrices. The author also conducts a lot of experiments showing that OATS is able to consistently outperform prior state-of-the-art across multiple benchmarks and compression rates while also improving on speed-up.

**Strengths:**

1.This paper propose a novel method for large transformers compression that utilizes the second moment of the input embeddings to approximate the model’s weight matrices as a sum of a sparse matrix and a low-rank matrix.

2.Extensive experiments on recent large language models demonstrate the effectiveness of the proposed method, which also generalizes well to vision Transformers.

**Weaknesses:**

Please see the questions.

**Questions:**

1. The author has clarified the differences between Wanda and OATS. However, an ablation study would further illustrate the impact of the low-rank term on performance. How much does the low-rank term contribute to the performance boost compared to Wanda alone?
2. The hardware speedup on GPU with N:M sparsity patterns can be shown and discussed in Section3.4.
3. The paper lacks details on how the sparsity pattern is defined for a matrix composed of a sparse matrix S plus a dense matrix LLL, especially within the context of N:M sparsity and structured pruning. Providing more detailed explanations would enhance clarity.
4. The baseline for models are all originally designed for LLMs. It would be valuable to compare against pruning metrics specifically designed for ViTs. This would present a stronger baseline to effectively showcase the proposed method’s advantages.
5. The motivation behind the choice of outlier information for the sparse term S is not entirely clear. Given that there are various methods for selecting the sparse components, such as using magnitude-based or gradient-based criteria, it would be helpful to know if the authors experimented with these or other selection methods.
6. Given that most pruning studies report results on 70B-parameter models, has the author tested their method on larger language models, such as Llama2-70B or the newer Llama3-70B?

---

> ### Author Response · Authors · 2024-11-17
>
> We would like to first thank the reviewer for spending the time to provide feedback, and engaging with the authors through several questions.
>
> >The author has clarified the differences between Wanda and OATS. However, an ablation study would further illustrate the impact of the low-rank term on performance. How much does the low-rank term contribute to the performance boost compared to Wanda alone?
>
> We thank the reviewer for the suggestion and we have added Appendix A.7 titled “Gap Between OATS and Wanda” highlighting the exact gaps for each performance metric between Wanda and OATS. This addition aims to quantify the impact of the low-rank term on performance. For the reviewer’s convenience, we have also included the relevant table below.
>
> | Model            | Compression | MMLU (↑) | Zero-Shot (↑) | Perplexity (↓) |
> |------------------|-------------|----------|---------------|----------------|
> | Phi-3 Mini       | 30%         | +1.21%   | +0.82%        | -0.44          |
> |                  | 40%         | +1.60%   | +1.24%        | -1.06          |
> |                  | 50%         | +5.42%   | +3.38%        | -2.05          |
> | Phi-3 Medium     | 30%         | +0.97%   | -0.01%        | -0.43          |
> |                  | 40%         | +1.65%   | +1.45%        | -0.79          |
> |                  | 50%         | +2.52%   | +2.43%        | -1.07          |
> | Llama-3 8B       | 30%         | +1.55%   | +0.71%        | +0.20          |
> |                  | 40%         | +2.13%   | +1.64%        | -0.50          |
> |                  | 50%         | +6.63%   | +2.44%        | -1.49          |
> | Llama-3 70B      | 30%         | -0.68%   | +0.05%        | -0.17          |
> |                  | 40%         | +0.73%   | +0.78%        | -0.40          |
> |                  | 50%         | +2.74%   | +0.45%        | -0.60          |
>
> > The hardware speedup on GPU with N:M sparsity patterns can be shown and discussed in Section3.4.
>
> The experiments conducted in Section 3.4 with N:M sparsity were designed as exploratory investigations to compare pruning algorithms with 2:4 sparsity against OATS under the same compression rate (but sparser N:M). Unfortunately, current NVIDIA GPUs only support 2:4 sparsity patterns, preventing us from benchmarking speed-ups on a GPU. However, consistent with prior works like [1,2], which also explore sparser N:M patterns, our goal is to demonstrate that models can achieve higher sparsity while maintaining performance. Through our results, we hope to incentivize companies to support sparser N:M patterns paving the way for greater speed-ups in the future.
>
> > The paper lacks details on how the sparsity pattern is defined for a matrix composed of a sparse matrix S plus a dense matrix LLL, especially within the context of N:M sparsity and structured pruning. Providing more detailed explanations would enhance clarity.
>
> We thank the author for highlighting a point of unclarity in our paper. Once we have the sparse plus low-rank decomposition, we are storing the compressed weight matrix as three separate matrices, a sparse matrix coinciding with the sparse term, and two matrices coinciding with the low-rank factorization of the low-rank term $LD^{-1}$. We have included the following sentence at the end of Section 2.3 of the manuscript to improve clarity:
>
> _The original weight matrix is replaced with three matrices: the sparse matrix $S D^{-1}$, and two matrices coinciding with the low-rank factorization of $LD^{-1}$._
>
> [1] Yin, L., Wu, Y., Zhang, Z., Hsieh, C.-Y., Wang, Y., Jia, Y., Pechenizkiy, M., Liang, Y., Wang, Z., & Liu, S. (2024). Outlier Weighed Layerwise Sparsity (OWL): A Missing Secret Sauce for Pruning LLMs to High Sparsity.
>
> [2] Sun, W., Zhou, A., Stuijk, S., Wijnhoven, R., Nelson, A. O., Li, H., & Corporaal, H. (2021). DominoSearch: Find layer-wise fine-grained N sparse schemes from dense neural networks.

---

> ### Author Response · Authors · 2024-11-17
>
> > The baseline for models are all originally designed for LLMs. It would be valuable to compare against pruning metrics specifically designed for ViTs. This would present a stronger baseline to effectively showcase the proposed method’s advantages.
>
> We appreciate the reviewer’s suggestion and we have added the following paragraph to Appendix A.1 directly addressing this issue:
>
> _There are a number of pruning approaches that have been specifically catered towards pruning vision transformers [1,2,3,4,5,6]. However, as much of the pruning literature developed on vision transformers involved models of much smaller scale than the large language models employed in this study, almost all of the prominent pruning algorithms require some form of training on the model parameters. As OATS was designed to require no training, OATS and the aforementioned pruning algorithms would not be comparable._
>
> > The motivation behind the choice of outlier information for the sparse term S is not entirely clear. Given that there are various methods for selecting the sparse components, such as using magnitude-based or gradient-based criteria, it would be helpful to know if the authors experimented with these or other selection methods.
>
> We share the same concerns as the reviewer that the choice of using the matrix $D$ may not be the optimal metric when determining the sparse term. We avoided utilizing gradient-based approaches as methods like OATS, DSNOT, SparseGPT, and Wanda can prune the layers through effectively a single forward pass. By requiring gradient information, it would increase the computational requirements of the pruning algorithm. In response to the reviewer, we have, however, extended our ablation studies in Section 3.3 to include experiments where the sparse term $S$ is determined via magnitude pruning instead. These results are presented in a new section Appendix A.5 titled “Magnitude-Based Pruning for the Sparse Component” which are included in the table below for the reviewer’s convenience:
>
> | Outlier Scaling        | MMLU (↑) | Zero-shot (↑) | Perplexity (↓) |
> |------------------------|----------|---------------|----------------|
> | Low-Rank Term Only     | 65.22    | **71.01**     | 12.49          |
> | Both Terms (OATS)      | **65.84** | 70.71         | **11.50**      |
>
> We have also included experiments in a new subsection Appendix A.3 titled “Using a Robust Scaling Matrix” where the scaling matrix $D$ utilizes a robust measurement of the second moments instead. We have included the new subsection below for the reviewer’s convenience:
>
> _To explore whether the scaling matrix $D$ is truly related to the outlier information, we run the following two experiments:_
> 1. _scaling by the square root of the features' second moments, as is currently done in OATS._
> 2. _scaling by the median of the features' absolute values (computed along batch and sequence dimensions):_
>
> $\qquad \qquad \qquad \qquad \qquad D_{robust} =$median$(|X|) $
>
> _The second experiment estimates the square root of the second moment of features in a manner that is robust (insensitive) to outliers akin to the Median Absolute Deviation estimator from the robust statistics literature [7]. The results of the two experiments are presented in the table below:_
>
> | Scaling Matrix   | MMLU (↑) | Zero-shot (↑) | Perplexity (↓) |
> |------------------|----------|---------------|----------------|
> | $D_{robust}$ | 55.54    | 65.77         | 18.59          |
> | $D$          | **59.99** | **68.41**     | **15.18**      |
>
> _The findings show that using the robust scaling method results in significantly worse performance. Hence, the scaling matrix $D$ that is sensitive to the outlier features and captures their scale leads to better compression._
>
> [1] Chen, T., Cheng, Y., Gan, Z., Yuan, L., Zhang, L., & Wang, Z. (2021). Chasing Sparsity in Vision Transformers: An End-to-End Exploration.
>
> [2] Yu, S., Chen, T., Shen, J., Yuan, H., Tan, J., Yang, S., Liu, J., & Wang, Z. (2022). Unified Visual Transformer Compression.
>
> [3] Yu, F., Huang, K., Wang, M., Cheng, Y., Chu, W., & Cui, L. (2022). Width and Depth Pruning for Vision Transformers.
>
> [4] Yu, L., & Xiang, W. (2023). X-Pruner: eXplainable Pruning for Vision Transformers.
>
> [5] Zhu, M., Han, K., Tang, Y., & Wang, Y. (2021). Visual Transformer Pruning.
>
> [6] Chavan, A., Shen, Z., Liu, Z., Liu, Z., Cheng, K.-T., & Xing, E. (2022). Vision Transformer Slimming: Multi-Dimension Searching in Continuous Optimization Space.
>
> [7] Huber, P. J. (1981). Robust statistics.

---

> ### Author Response · Authors · 2024-11-17
>
> > Given that most pruning studies report results on 70B-parameter models, has the author tested their method on larger language models, such as Llama2-70B or the newer Llama3-70B?
>
> Thank you for this great suggestion. In response, we have run Llama-3 70B experiments for OATS and the pruning benchmarks and have updated the manuscript to include those experiments. These results are included in the table below for your convenience:
>
> | Compression | Method      | MMLU (↑)       | Zero-Shot (↑)       | Perplexity (↓)       |
> |-------------|-------------|----------------|----------------------|-----------------------|
> | 0%          | Dense       | 79.63          | 75.27               | 2.68                 |
> | 30%         | SparseGPT   | 78.28          | 75.07               | 3.24                 |
> |             | Wanda       | **79.15**          | 75.19               | 3.28                 |
> |             | DSNoT       | 79.00          | **75.54**               | 3.27                 |
> |             | OATS        | 78.47          | 75.24               | **3.07**                 |
> | 40%         | SparseGPT   | 76.29          | 74.63               | 3.99                 |
> |             | Wanda       | 77.16          | 74.10               | 4.08                 |
> |             | DSNoT       | 77.70          | 74.29               | 4.10                 |
> |             | OATS        | **77.89**          | **74.88**               | **3.68**                 |
> | 50%         | SparseGPT   | 72.47          | 73.17               | 5.27                 |
> |             | Wanda       | 72.04          | 72.85               | 5.38                 |
> |             | DSNoT       | 72.76          | 72.91               | 5.58                 |
> |             | OATS        | **74.79**          | **73.30**               | **4.78**                 |
>
> >
>
> We would like to thank the reviewer again for asking many insightful questions that have led to the improvement of our work. Should the reviewer find it fitting, we would be appreciative of any potential reconsideration of the score.

---

> > ### Comment · Reviewer_rXxS · 2024-11-21
> >
> > Dear authors,
> >
> > Thank you for considering my comments and for the well-prepared rebuttal. I will keep my original positive score.
> >
> > Best wishes,

---

> > > ### Author Response · Authors · 2024-12-04
> > > **Thank You**
> > >
> > > Dear Reviewer rXsS,
> > >
> > > As the discussion period draws to a close, we would like to sincerely thank the reviewer for their participation during the discussion, for complimenting on the preparedness of our rebuttal, and for their overall positive assessment of our work.
> > >
> > > Best,
> > >
> > > The Authors.

---

### Official Review · Reviewer_PLVG · 2024-10-29

**Soundness:** 4
**Presentation:** 4
**Contribution:** 3
**Rating:** 8
**Confidence:** 4

**Summary:**

This paper proposes a relatively novel model compression technique aimed at Transformer architectures, in which weight matrices are approximated as a sum of a sparse and a low-rank matrix. To control for the (previously documented) outlier feature problem, weights are also scaled by the second moment of their input.

**Strengths:**

The method proposed by this paper is well-explained and well-justified. The actual practical algorithm is easy to follow.

Real-time speedup is shown in the CPU setting.

The experiments cover a range of model sizes (3.8-14B parameters). I especially liked seeing results on fairly small models, since those may be harder to compress.

Section 5 is an interesting way to look at the problem, which I think can lead to interesting further work in the direction of interpretability.

**Weaknesses:**

The choice of the rank ratio parameter could have been better explored (in particular, looking at multiple architectures/tasks).

Typos:
Line 18: “approximating each weight” -> “approximating each weight matrix”
Line 142: “the activations are calculated through a calibration set that is propagated through the compressed layers” - should be uncompressed?

**Questions:**

It would be nice to see a bigger hyperparameter selection section with more architerctures considered.

---

> ### Author Response · Authors · 2024-11-17
>
> We would first like to thank the reviewer for providing us with valuable feedback and for highlighting concerns with regard to typos:
>
> > Line 18: “approximating each weight” -> “approximating each weight matrix”
>
> Thank you for pointing this out, we have since updated the abstract to correct this typo.
>
> >Line 142: “the activations are calculated through a calibration set that is propagated through the compressed layers” - should be uncompressed?
>
> This is actually not a typo and the choice of calculating the input activations through the compressed layers was a deliberate decision based on prior pruning algorithms that did the same (SparseGPT, Wanda, and DSNOT).
>
> > It would be nice to see a bigger hyperparameter selection section…
>
> To provide a better picture of how the hyperparameters impact OATS, we have included the results for additional configurations for the Llama-3 8B model and the Phi-3 Mini model in Appendix A.6 titled “Additional Hyperparameter Tests for OATS”. We have included the table below for the reviewer’s convenience:
>
> | Model          | Compression | Rank Ratio | MMLU (↑) | Zero-Shot (↑) | Perplexity (↓) |
> |----------------|-------------|------------|----------|---------------|----------------|
> | Phi-3 Mini     | 30%         | 0.1        | 68.70    | 71.65         | 10.24          |
> |                |             | 0.2        | 68.02    | 71.81         | 10.21          |
> |                |             | 0.3        | 69.28    | 72.07         | 10.28          |
> |                | 40%         | 0.1        | 65.75    | 69.94         | 11.57          |
> |                |             | 0.2        | 65.84    | 70.71         | 11.50          |
> |                |             | 0.3        | 66.81    | 70.54         | 11.60          |
> |                | 50%         | 0.1        | 57.96    | 67.37         | 15.48          |
> |                |             | 0.2        | 59.12    | 68.02         | 15.13          |
> |                |             | 0.3        | 58.68    | 68.63         | 15.47          |
> | Llama-3 8B     | 30%         | 0.1        | 63.62    | 68.99         | 9.35           |
> |                |             | 0.2        | 63.09    | 69.54         | 9.09           |
> |                | 40%         | 0.1        | 61.44    | 68.23         | 9.23           |
> |                |             | 0.2        | 61.97    | 68.43         | 9.09           |
> |                | 50%         | 0.1        | 56.46    | 65.33         | 10.85          |
> |                |             | 0.2        | 56.07    | 65.51         | 10.70          |
>
> Unfortunately, due to computational constraints, we were unable to perform a more extensive grid search over the hyperparameters for each model. However, given that all Phi-3 experiments utilized a rank ratio of 0.25 and all Llama-3 experiments utilized a rank ratio of 0.3, we believe that this is a testament to the robustness of OATS' performance to its hyperparameters.
>
> > with more architerctures considered.
>
> We agree with the reviewer and we also believe that our experiments would benefit from including larger models. Thus, we have run experiments on Llama-3 70B for OATS and its pruning benchmarks, which we have included in the manuscript and below for the reviewer’s convenience:
>
> | Compression | Method      | MMLU (↑)       | Zero-Shot (↑)       | Perplexity (↓)       |
> |-------------|-------------|----------------|----------------------|-----------------------|
> | 0%          | Dense       | 79.63          | 75.27               | 2.68                 |
> | 30%         | SparseGPT   | 78.28          | 75.07               | 3.24                 |
> |             | Wanda       | **79.15**          | 75.19               | 3.28                 |
> |             | DSNoT       | 79.00          | **75.54**               | 3.27                 |
> |             | OATS        | 78.47          | 75.24               | **3.07**                 |
> | 40%         | SparseGPT   | 76.29          | 74.63               | 3.99                 |
> |             | Wanda       | 77.16          | 74.10               | 4.08                 |
> |             | DSNoT       | 77.70          | 74.29               | 4.10                 |
> |             | OATS        | **77.89**          | **74.88**               | **3.68**                 |
> | 50%         | SparseGPT   | 72.47          | 73.17               | 5.27                 |
> |             | Wanda       | 72.04          | 72.85               | 5.38                 |
> |             | DSNoT       | 72.76          | 72.91               | 5.58                 |
> |             | OATS        | **74.79**          | **73.30**               | **4.78**                 |
>
> >
>
> We again would like to sincerely thank the reviewer for their comments and we remain available for discussion should the reviewer have any more inquiries prior to the discussion deadline. If the reviewer finds appropriate, the authors would be appreciative of any further reconsiderations of the score.

---

> > ### Comment · Reviewer_PLVG · 2024-11-18
> > **Thank you for the response.**
> >
> > I thank the reviewers for the clarification and additional experiments, and agree with the analysis that the hyperparameter search results provide evidence for the method's robustness.
> >
> > Regarding Line 142 (compressed vs uncompressed weights in computing inputs). Then it would be helpful to clarify this in Algorithm 2.

---

> > > ### Author Response · Authors · 2024-11-21
> > >
> > > Thank you for the additional suggestion of editing the Algorithm 2 bubble. We have now added *Layer Inputs Propagated through Prior Compressed Layers* to the algorithm bubble when describing the layer inputs that are used to calculate the scaling matrix $D$.
> > >
> > > > I especially liked seeing results on fairly small models, since those may be harder to compress…It would be nice to see… more architerctures considered.
> > >
> > > We wanted to add that we have since run additional experiments on the Qwen 2.5 3B Instruct model to provide an even better understanding of how OATS performs on a wide range of different LLM architectures. We have included the results in a new section, Appendix A.8, titled *Qwen 2.5 Experiments* and included the table below for the reviewer’s convenience:
> > >
> > > | Compression | Method      | MMLU (↑)       | Zero-Shot (↑)       | Perplexity (↓)       |
> > > |-------------|-------------|----------------|----------------------|-----------------------|
> > > | 0%          | Dense       | 65.99          | 68.49               | 11.02                 |
> > > | 30%         | SparseGPT   | **65.65**          | 67.91               | 11.55                 |
> > > |             | Wanda       | 65.46          | 68.08               | 11.66                 |
> > > |             | DSNoT       | 65.65          | 68.21              | 11.67                 |
> > > |             | OATS        | 65.36          | **68.74**               | **11.45**                 |
> > > | 40%         | SparseGPT   | 63.04          | 67.64               | 12.56                 |
> > > |             | Wanda       | 61.88          | 67.14               | 12.89                 |
> > > |             | DSNoT       | 62.26          | 67.42               | 12.91                 |
> > > |             | OATS        | **64.30**          | **68.76**               | **12.31**                 |
> > > | 50%         | SparseGPT   | 57.43          | 64.36               | 14.92                 |
> > > |             | Wanda       | 55.39          | 64.10               | 16.27                 |
> > > |             | DSNoT       | 55.78          | 64.77               | 16.43                 |
> > > |             | OATS        | **58.78**          | **65.74**               | **14.91**                 |
> > >
> > > We thank the reviewer again for their continued engagement with us. We remain available for discussion should the reviewer have any other suggestions that would lead to improvements for our work and its score.

---

> ### Author Response · Authors · 2024-12-04
> **Thank You**
>
> Dear Reviewer PLVG,
>
> As the discussion period wraps up, we would like to offer our final thanks to the reviewer for responding to our comments, for providing additional suggestions beyond their initial review, and for their positive evaluation of our work.
>
> Best,
>
> The Authors.

---

### Official Review · Reviewer_BUra · 2024-11-01

**Soundness:** 2
**Presentation:** 2
**Contribution:** 1
**Rating:** 3
**Confidence:** 5

**Summary:**

This paper introduces OATS, a novel compression technique for large-scale transformers that combines sparse and low-rank approximations to reduce memory and compute costs without requiring costly retraining. OATS scales model weights by the second moment of input embeddings to preserve essential outlier features in transformers, ensuring model performance is maintained during compression. This approach addresses the typical performance degradation seen with increasing compression in existing pruning methods.

**Strengths:**

+ Low-rankness plus sparsity is a good fit for compressing LLMs without retraining.
+ By separating the model into a sparse and a low-rank part, the approximation error can be theoretically reduced as the two parts can compensate for each other.
+ This paper provides measurements for practical speedups on CPUs with existing sparse computation frameworks.

**Weaknesses:**

- The novelty is limited. The combination of low-rankness and sparsity is an old topic that has been explored for many years [R1, R2]. Applying the well-established approximation techniques to decompose/compress the large matrices in LLMs has little technical contribution. Besides, compressing DNN models using low-rank and sparse decomposition has already been well explored in [R3]. This paper just scales it to larger models and matrices. Authors are encouraged to specify the unique difference from existing approaches and why this difference is also unique for LLMs.
- The proposed Truncated SVD and Threshold strategies to achieve low-rankness and sparsity are too trivial. It is unknown how to decide the rank and number of zeroes. Besides, the order of applying SVD and thresholding has a significant impact on the approximation errors. Authors are encouraged to clearly explain why using such decomposition strategies.
- This paper claims "outlier information" in this paper. However, I have not seen any analysis or explanation for the "outlier information," and the proposed solution is not related to the "outlier information." Instead, this paper seems to directly apply the pruning approaches proposed in Wanda. The authors are encouraged to provide explanations of why the diagonal matrix is related to "outlier information" and why it is good for compression.
- Many works have been proposed to compress LLMs with low-rankness and sparsity [R4-R6]. The authors have not presented the main differences among them and the unique contributions that stand out from those works.
- Even though theoretical analysis may not have a practical guarantee of accuracy, the authors are encouraged to provide.
- The paper presentation could be improved, especially the math equations.

[R1] Sparse and Low-Rank Matrix Decompositions, Forty-Seventh Annual Allerton Conference, 2009.

[R2] Godec: Randomized low-rank & sparse matrix decomposition in noisy case. ICML 2011.

[R3] On compressing deep models by low rank and sparse decomposition, CVPR 2017.

[R4] Slope: Double-pruned sparse plus lazy low-rank adapter pretraining of llms

[R5] LoSparse: Structured Compression of Large Language Models based on Low-Rank and Sparse Approximation

[R6] SLiM - One-shot Quantized Sparse Plus Low-rank Approximation of LLMs

**Questions:**

See the Weakness. Additionally, why use inverse transformation to reach the compressed weight? What is the actual speedup when using N:M sparsity on GPUs.

---

> ### Author Response · Authors · 2024-11-17
> **OATS Novelty and Differences**
>
> ## **OATS Novelty and Differences (Part 1/5)** ##
>
> We appreciate the reviewer’s critical assessment of our paper and we will attempt to address each of the reviewer’s concerns.
>
> >**[Reviewer Comment]: The combination of low-rankness and sparsity is an old topic that has been explored for many years [R1, R2].**
>
> We apologize for having missed seminal work [1,2] during our literature search and have fixed the manuscript appropriately to include citing the former when Robust PCA is presented in the paper, and the latter when presenting the alternating thresholding algorithm.  **We want to clarify that the novelty associated with our work is the effectiveness of Robust PCA for compressing large transformer models once an outlier scaling is applied -- not the Robust PCA problem itself.** When introducing the Robust PCA problem, we did cite two seminal works on the problem, a more widely cited work by the same authors of [1] titled _Rank-Sparsity Incoherence for Matrix Decomposition_ and a work by Candes et al. titled _Robust principal component analysis?_ (2009).
>
> > **[Reviewer Comment]: Besides, compressing DNN models using low-rank and sparse decomposition has already been well explored in [R3]. This paper just scales it to larger models and matrices.  Authors are encouraged to specify the unique difference from existing approaches and why this difference is also unique for LLMs.**
>
> We again apologize for not having cited this important and relevant work and we thank the reviewer for pointing it out. To rectify our mistake, we have now added the following paragraph to Appendix A.1, providing an in-depth explanation of the differences between the two algorithms and why such differences are unique to LLMs:
>
> **[New Section | Page 18 ]:** _[3] introduced a method for sparse and low-rank decomposition of CNNs, including AlexNet and GoogLeNet, by solving the following optimization problem:_
>
> $$ \min_{S, L \in \mathbb{R}^{d_{out}\times d_{in}}} ||Y - (S+L)X||_2^{2} \\; \text{s.t.} \\; ||W - (S + L) ||_F^2\leq \gamma, \text{Rank}(L)\leq r, ||S||_0 \leq k $$
>
> _where $Y = W X$. In contrast, OATS employs a different approach, solving:_
>
> $$\min_{S, L \in \mathbb{R}^{d_{out} \times d_{in}}} || W - S - L ||_F^2 \\; \text{ s.t. } \\; \text{Rank}(L) \leq r, \\; ||S||_0 \leq k. $$
>
> _**A key distinction between these methods lies in their objectives: the former directly minimizes reconstruction error, while OATS adopts a simpler formulation.** One might question why not follow the approach of minimizing reconstruction error. **As noted in DSNoT [4], pruning methods that prioritize minimizing reconstruction error can degrade model performance in large transformers, particularly in the presence of outlier features.** Their findings highlight the importance of avoiding pruning weights within outlier channels. **Since feature outliers are a phenomenon unique to large transformer models [5], this issue would not have been relevant to the work of [3], which predates the transformer era.**_
>
> > **[Reviewer Comment]: The proposed Truncated SVD and Threshold strategies to achieve low-rankness and sparsity are too trivial.**
>
> We acknowledge the reviewer's point that OATS is a simple algorithm. However, this simplicity is precisely what makes OATS appealing. **It demonstrates that a conceptually simple approach, which has not been widely applied to large-scale transformer models, can outperform more complex pruning methods.**
>
> [1] Chandrasekaran, V., Sanghavi, S., Parrilo, P. A., & Willsky, A. S. (2009). Sparse and Low-Rank Matrix Decompositions.
>
> [2] Zhou, T., & Tao, D. (2011). GoDec: randomized low-rank & sparse matrix decomposition in noisy case.
>
> [3] Yu, X., Liu, T., Wang, X., & Tao, D. (2017). On Compressing Deep Models by Low Rank and Sparse Decomposition.
>
> [4] Zhang, Y., Zhao, L., Lin, M., Yunyun, S., Yao, Y., Han, X., Tanner, J., Liu, S., & Ji, R. (2024). Dynamic Sparse No Training: Training-Free Fine-tuning for Sparse LLMs.
>
> [5] Dettmers, T., Lewis, M., Belkada, Y., & Zettlemoyer, L. (2024). LLM.int8(): 8-bit matrix multiplication for transformers at scale.

---

> ### Author Response · Authors · 2024-11-17
> **Additional Ablations for the OATS Algorithm**
>
> ## **Additional Ablations for the OATS Algorithm (Part 2/5)** ##
>
> > **[Reviewer Comment]: It is unknown how to decide the rank and number of zeroes.**
>
> All Llama-3 experiments use a rank ratio of 0.3, and all Phi-3 experiments use a rank ratio of 0.25, demonstrating the robustness of OATS to the rank ratio parameter.
>
> A related and valid question is why the same rank ratio can be applied across different layers of the model. **We hypothesize that the weight matrices possess meaningful low-rank subspaces of roughly similar dimensions across layers. While we are actively pursuing a theoretical investigation to better understand this phenomenon, we believe that such an analysis falls outside the scope of the current paper. Evidence for the existence of a meaningful low-rank subspace is presented in a concurrent submission to the ICLR conference [1],** which examines the spectral properties of weight matrices through the lens of Random Matrix Theory. Specifically, the study demonstrates that the spectrum includes a small number of outlier eigenvalues, whose removal significantly degrades performance.
>
> > **[Reviewer Comment]: Besides, the order of applying SVD and thresholding has a significant impact on the approximation errors.**
>
> We thank the reviewer for providing a thought-provoking remark about whether the order utilized by OATS is the optimal order to be used. To address this question, we have run additional experiments and added the following subsection in Appendix A.4 titled “Switching the Order of Thresholding”:
>
> **[New Section | Page 20]:** _OATS opts to perform the singular-value thresholding first followed by the hard thresholding similar to [2]. However, one might consider whether the alternative order could lead to faster convergence or a better approximation. Presented in the table below is an extension of the ablation studies presented in Section 3.3, reporting the performance of OATS where the hard-thresholding is performed first:_
>
> | First Thresholding Operation         | MMLU (↑) | Zero-shot (↑) | Perplexity (↓) |
> |--------------------------------------|----------|---------------|----------------|
> | Hard-Thresholding                    | 65.51    | 70.54         | 11.72          |
> | Singular Value Thresholding (OATS)   | **65.84** | **70.71**     | **11.50**      |
>
> _**While the performance still remains competitive, across all performance metrics, the switched order falls short of matching the original order presented in the Algorithm.**_
>
> > **[Reviewer Comment]: This paper claims "outlier information" in this paper. However, I have not seen any analysis or explanation for the "outlier information," and the proposed solution is not related to the "outlier information" … The authors are encouraged to provide explanations of why the diagonal matrix is related to "outlier information" and why it is good for compression."**
>
> We would like to thank the reviewer for this wonderful inquiry. In response, we have included a new subsection titled “Using a Robust Scaling Matrix” to Appendix A.3, which includes additional experiments exploring whether the matrix $D$ is truly capturing the outlier information, and if that is the reason why it is good for compression:
>
> **[New Section | Page 20]:** _To explore whether the scaling matrix $D$ is truly related to the outlier information, we run the following two experiments:_
> 1. _scaling by the square root of the features' second moments, as is currently done in OATS._
> 2. _scaling by the median of the features' absolute values (computed along batch and sequence dimensions):_
>
> $\qquad \qquad \qquad \qquad \qquad D\_{robust} =$median$(|X|) $
>
> _The second experiment estimates the square root of the second moment of features in a manner that is **robust (insensitive) to outliers** akin to the Median Absolute Deviation estimator from the robust statistics literature [3]. The results of the two experiments are presented in the table below:_
>
> | Scaling Matrix   | MMLU (↑) | Zero-shot (↑) | Perplexity (↓) |
> |------------------|----------|---------------|----------------|
> | $D_{robust}$ | 55.54    | 65.77         | 18.59          |
> | $D$          | **59.99** | **68.41**     | **15.18**      |
>
> _**The findings show that using the robust scaling method results in significantly worse performance. Hence, the scaling matrix $D$ that is sensitive to the outlier features and captures their scale leads to better compression.**_
>
> The reason why the scaling is good for compression has also been touched on in prior works like DSNoT and Wanda. Both papers utilize the same scaling matrix as OATS and both reasoned that the scaling is needed to steer the algorithm away from pruning weights that are in an outlier channel.
>
> [1] https://openreview.net/forum?id=MmWkNmeDNE
>
> [2]  Zhou, T., & Tao, D. (2011). GoDec: randomized low-rank & sparse matrix decomposition in noisy case.
>
> [3] Huber, P. J. (1981). Robust statistics.

---

> ### Author Response · Authors · 2024-11-17
> **Similarity with Wanda**
>
> ## **Similarity with Wanda (Part 3/5)** ##
>
> > **[Reviewer Comment]: Instead, this paper seems to directly apply the pruning approaches proposed in Wanda.**
>
> Thank you for this comment. **We wish to clarify that the Wanda algorithm is strictly for pruning dense weight matrices into sparse matrices. Unlike OATS, there is no sparse and low-rank decomposition or alternating thresholding being done.** Throughout the paper, we have emphasized that the scaling done by OATS is inspired by Wanda, with an additional paragraph in the Related Works section specifically dedicated to highlighting that OATS would reduce to Wanda, in the specific case where the rank ratio is 0.
>
> To further illustrate the differences between OATS and Wanda, we have included a new section Appendix A.7 titled “Performance Gap Between OATS and Wanda”, that highlights the exact gap between the two compression approaches for each performance metric. We have included the table below for the reviewer’s convenience:
>
> | Model            | Compression | MMLU (↑) | Zero-Shot (↑) | Perplexity (↓) |
> |------------------|-------------|----------|---------------|----------------|
> | Phi-3 Mini       | 30%         | +1.21%   | +0.82%        | -0.44          |
> |                  | 40%         | +1.60%   | +1.24%        | -1.06          |
> |                  | 50%         | +5.42%   | +3.38%        | -2.05          |
> | Phi-3 Medium     | 30%         | +0.97%   | -0.01%        | -0.43          |
> |                  | 40%         | +1.65%   | +1.45%        | -0.79          |
> |                  | 50%         | +2.52%   | +2.43%        | -1.07          |
> | Llama-3 8B       | 30%         | +1.55%   | +0.71%        | +0.20          |
> |                  | 40%         | +2.13%   | +1.64%        | -0.50          |
> |                  | 50%         | +6.63%   | +2.44%        | -1.49          |
> | Llama-3 70B      | 30%         | -0.68%   | +0.05%        | -0.17          |
> |                  | 40%         | +0.73%   | +0.78%        | -0.40          |
> |                  | 50%         | +2.74%   | +0.45%        | -0.60          |

---

> ### Author Response · Authors · 2024-11-17
> **Additional Related Works and Overall Presentation**
>
> ## **Additional Related Works and Overall Presentation (Part 4/5)** ##
> > **[Reviewer's Comment]: Many works have been proposed to compress LLMs with low-rankness and sparsity [R4-R6]. The authors have not presented the main differences among them and the unique contributions that stand out from those works.**
>
> We sincerely apologize for the oversight in not citing important and relevant works [R4, R6], and we greatly appreciate the reviewer for bringing these to our attention. We have corrected this error, and the following paragraphs outline the additions and adjustments made to address it.
>
> To highlight and compare [R4] with OATS, we have included the following paragraph in Appendix A.1:
>
> **[New Section | Page 19]:** _In [1], the authors propose SLOPE, a novel method for accelerating the pre-training phase of LLMs by incorporating N:M sparsity and adding low-rank components to the model weights to enhance model capacity. Similar to OATS, SLOPE leads to a sparse plus low-rank structure in the model’s weight matrices, **however, the low-rank terms are introduced during the final phase of pre-training and are actively trained on the model loss function.** In contrast, **OATS is designed as a lightweight method to accelerate inference. OATS does not require any training or fine-tuning**, but instead approximates pre-trained weight matrices by solving the Robust PCA problem._
>
> For [R6], we thank the reviewers for bringing this concurrent work to our attention. We were, unfortunately, unable to cite the paper since it was posted publicly on ArXiV only after the submission deadline for ICLR. We have now included the following paragraph in Appendix A.1 to highlight its differences with OATS:
>
> **[New Section | Page 19]:** _An independent and concurrent work with OATS proposes SLIM [2], a novel pipeline that combines pruning and quantization. To restore lost performance from compression, SLIM derives a low-rank term using singular-value thresholding and adopts a scaling technique akin to OATS. However, instead of the $L^2$ norm, SLIM utilizes the average absolute value across the batch and sequence dimensions. As a further deviation from OATS, **SLIM is also not performing an alternating thresholding algorithm. Instead, they perform a single quantization and pruning step to initialize the quantized and sparse term, followed by a single singular value thresholding step to establish the low-rank term.**_
>
> Regarding [R5], **we previously cited it under the “Structured Pruning and Low-Rank Adaptation” paragraph in the Related Works section**. We have included the paragraph below for the reviewer’s convenience:
>
> **[Page 10]:**  _Recent works, such as **LoSparse**, LoRAPrune, and APT, propose variations of applying structured pruning on the weights while incorporating a low-rank adapter that is trained via gradient descent. These are markedly different than OATS, which does not employ any fine-tuning with low-rank adapters, nor does it perform structured pruning (but rather a sparse plus low-rank decomposition which can be thought of as a combination of structured and unstructured pruning)._
>
> >**[Reviewer's Comment]: The paper presentation could be improved, especially the math equations.**
>
> We thank the reviewer for raising their concerns and we have provided an additional annotation to the following equation in Section 2.4 when defining the rank-ratio, $\kappa = \frac{r(d_{out} + d_{in})}{(1-\rho)d_{out} \cdot d_{in}}$, clarifying that the numerator represents the number of parameters in the low-rank term and that the denominator represents the total number of nonzero parameters in the compressed layer.
>
> > **[Reviewer's Comment]: Additionally, why use inverse transformation to reach the compressed weight?**
>
> We thank the reviewer for raising a potential point of unclarity in our work. In OATS, the alternating thresholding algorithm returns a sparse plus low-rank decomposition that approximates $L+S \approx W D$. From this equation, if one wanted a sparse plus low-rank approximation of the original weight matrix, they would need to multiply by $D^{-1}$ on the right. To improve clarity, we have edited lines 136-137 to instead read:
>
> _…which gives a sparse plus low-rank approximation of $WD \approx S+ L$. OATS then applies the inverse transformation to reach the final compressed weight:_
>
> $$ W_{compressed} := (L + S) D^{-1}.$$
>
> [1] Mozaffari, M., Yazdanbakhsh, A., Zhang, Z., & Mehri Dehnavi, M. (2024). SLoPe: Double-Pruned Sparse Plus Lazy Low-Rank Adapter Pretraining of LLMs.
>
> [2] Mozaffari, M., & Mehri Dehnavi, M. (2024). SLiM: One-shot Quantized Sparse Plus Low-rank Approximation of LLMs.

---

> ### Author Response · Authors · 2024-11-17
> **N:M Speed-Up**
>
> ## **N:M Speed-Up (Part 5/5)** ##
>
> > **[Reviewer's Comment]: What is the actual speedup when using N:M sparsity on GPUs.**
>
> The experiments conducted in Section 3.4 with N:M sparsity were designed as exploratory investigations to compare pruning algorithms with 2:4 sparsity against OATS under the same compression rate (but sparser N:M). Unfortunately, current NVIDIA GPUs only support 2:4 sparsity patterns, preventing us from benchmarking speed-ups on a GPU. However, consistent with prior works like [1, 2], which also explore sparser N:M patterns, our goal is to demonstrate that models can achieve higher sparsity while maintaining performance. Through our results, we hope to incentivize companies to support sparser N:M patterns paving the way for greater speed-ups in the future.
>
> [1] Yin, L., Wu, Y., Zhang, Z., Hsieh, C.-Y., Wang, Y., Jia, Y., Pechenizkiy, M., Liang, Y., Wang, Z., & Liu, S. (2024). Outlier Weighed Layerwise Sparsity (OWL): A Missing Secret Sauce for Pruning LLMs to High Sparsity.
>
> [2] Sun, W., Zhou, A., Stuijk, S., Wijnhoven, R., Nelson, A. O., Li, H., & Corporaal, H. (2021). DominoSearch: Find layer-wise fine-grained N sparse schemes from dense neural networks.
>
> >
>
> We would like to again thank the reviewer for deeply engaging with our paper and for providing numerous valuable comments that have allowed us to improve the paper. Given the comments and additions to the paper, should the reviewer find it appropriate, we would be grateful for any potential reconsideration of our score.

---

> ### Author Response · Authors · 2024-11-22
>
> Dear reviewer, we would greatly appreciate it if you could take a moment to read our rebuttal and provide any further feedback to the improvements made in response to your comments. This would give us enough time to respond should there be any further changes that the reviewer thinks are needed to improve the paper and its score. We understand the demanding nature of the review process and appreciate the time and effort that the reviewer is dedicating to this task.

---

> ### Author Response · Authors · 2024-11-26
>
> Dear reviewer, we apologize for the repeated emails, however, as the deadline for editing the manuscript is tomorrow, we wanted to ensure that all the reviewer’s concerns have been addressed, especially given that their initial review was quite critical compared to the others. We would greatly appreciate it if the reviewer could provide us with some feedback so that we could take further action should the reviewer have any more concerns prior to the deadline. We want to thank the reviewer again for their in-depth review and reiterate that we do very much appreciate the comments and suggestions that were made in the original review.

---

> ### Author Response · Authors · 2024-11-29
>
> Dear reviewer BUra, as we await your reply, we would like to bring to your attention that we have re-formatted, added a few clarifying sentences, and bolded key sentences in our replies addressing the reviewer’s comments. We hope that these changes make our responses easier to read for the reviewer. Please do not hesitate to reach out on the discussion page if there is anything that we, the authors, can do to ameliorate the reviewer’s process of reading/replying to our comments. We thank the reviewer and look forward to hopefully hearing back from them before the discussion deadline.

---

### Official Review · Reviewer_Gg5U · 2024-11-05

**Soundness:** 3
**Presentation:** 4
**Contribution:** 3
**Rating:** 8
**Confidence:** 3

**Summary:**

The paper introduces OATS (Outlier-Aware Pruning Through Sparse and Low-Rank Decomposition), a method designed to compress large transformer models without the need for retraining. The central concept involves representing weight matrices as the sum of a sparse and low-rank matrix.

The authors propose an iterative alternating thresholding technique to compute the joint sparse and low-rank decomposition of a matrix. They also focus on preserving outliers by scaling weights according to input embeddings prior to decomposition. The results indicate that OATS performs effectively on both language and vision tasks, outperforming other pruning methods proposed for transformers across various compression levels and tasks. Additionally, OATS offers speed improvements on the CPU.

**Strengths:**

- The concept of compressing a model as a sum of a sparse and low-rank matrix is very promising. Unlike most prior methods, which focus on one approach, OATS leverages both to potentially enhance performance.

- OATS is retraining-free, which is crucial for practical applications where even a single backpropagation pass can be computationally prohibitive.

- The framework has been tested on state-of-the-art models like Llama and ViT, demonstrating competitive performance.

- The Alternating Thresholding technique in OATS heuristically finds an effective combination of low-rank and sparse components and accommodates different sparsity patterns.

- By scaling the weight matrix $W$ with a diagonal rescaling matrix $D$, OATS emphasizes outliers, enabling outlier-aware compression that avoids performance degradation.

**Weaknesses:**

One concern is that the method relies on multiple calls to truncated SVD, which can be computationally intensive. Specifically, finding the top-$r$ singular values of an $m \times n$ matrix has a time complexity of $O(mnr)$. Given that compression speed is a significant factor for practical applications, it would be helpful if the authors could clarify the time complexity and wall-clock time spent on the compression process of the overall algorithm. This would offer a more concrete understanding of its practicality.

**Questions:**

One potential extension of this work could involve incorporating quantization into the proposed framework. Although this addition may be a long shot, integrating quantization could make the model a unified approach to transformer compression. Could the authors provide insights on whether quantization can be integrated with their current framework, or if not, what are the main challenges?

---

> ### Author Response · Authors · 2024-11-17
>
> The authors would like to extend their gratitude to the reviewer for highlighting the strengths of our paper, raising valid concerns about the runtime of OATS, and inquiring about follow-ups in regards to extending the algorithm to the quantization setting.
>
> >Given that compression speed is a significant factor for practical applications, it would be helpful if the authors could clarify the time complexity and wall-clock time spent on the compression process of the overall algorithm. This would offer a more concrete understanding of its practicality.
>
> We thank the reviewer for the suggestion and in response have included a new section Appendix A.2 titled ”Time Complexity and Wall-Clock Time for OATS” which we have included below for the reviewer’s convenience:
>
> _The time complexity for OATS is $\mathcal{O}(LN\alpha)$ where $L$ is the number of transformer blocks, $N$ is number of iterations, and_
>
> $\alpha =$ max$\_W d^{W}\_{out} \cdot d^{W}\_{in} \cdot r^{W}$
>
> _where the max is taken over the weight matrices, $W \in \mathbb{R}^{d^{W}\_{out} \times d^{W}\_{in}}$, in a transformer block and $r^{W}$ is the rank of the low-rank term for that weight matrix. The value $\alpha$ represents the time complexity needed to perform the singular value thresholding in OATS._
>
> _The table below reports the wall-clock time (in seconds) needed to perform a single iteration of the alternating threshold algorithm for a single transformer block for the different models that were compressed. All experiments utilized a single NVIDIA A40 with 48GB of GPU memory._
>
> | Phi-3 Mini (3.8B) | Phi-3 Medium (14B) | Llama-3 8B | Llama-3 70B |
> |--------------------|--------------------|------------|-------------|
> | 8.85              | 26.02              | 17.10      | 152.80      |
>
> _While OATS does require more wall-clock time than prior pruning algorithms, in practice, model compression would only need to be performed once before deployment. This trade-off is therefore worthwhile given the substantial performance improvements, particularly on more challenging tasks like MMLU. Furthermore, like prior pruning algorithms, compressing the layers within a single transformer block can be done in parallel. For example, the time needed per transformer block of Llama-3 70B can be reduced to 71.10 seconds by compressing in parallel across four NVIDIA A40 GPUs._
>
> _The total wall-clock time can also be reduced by lowering the number of OATS iterations. Presented in the Table below is an exploratory experiment compressing Llama-3 70B by 50\% with a rank ratio of 0.3 with only 20 iterations. Even with only a quarter of the iterations, OATS is still able to outperform all prior pruning algorithms across all performance metrics._
>
> | MMLU (↑) | Zero-shot (↑) | Perplexity (↓) |
> |----------|---------------|----------------|
> | 74.02    | 73.41         | 4.95           |
>
>
> > One potential extension of this work could involve incorporating quantization into the proposed framework. Although this addition may be a long shot, integrating quantization could make the model a unified approach to transformer compression. Could the authors provide insights on whether quantization can be integrated with their current framework, or if not, what are the main challenges?
>
> We share the same interest as the reviewer in integrating quantization with OATS. The formulation proposed in our work is flexible and can indeed accommodate that. However, one of the associated challenges is deciding which terms in the sparse and low-rank decomposition should be quantized. Depending on that decision, one would obtain one of the following three optimization problems:
>
> 1. min$\_{S\_Q, L\_Q} ||S\_Q + L\_Q - W|| \text{ s.t. } ||S\_Q||\_0 \leq k, S\_Q$ is quantized, rank$(L\_Q)\leq r, L\_Q$ is quantized
>
> 2. min$\_{S\_Q, L} \\; \\; ||S\_Q + L - W|| \\; \\; \text{ s.t. } ||S\_Q||\_0 \leq k, S\_Q$ is quantized, rank$(L)\leq r$
>
> 3. min$_{S, L\_Q} \\; \\; ||S + L\_Q - W||  \\; \\; \text{ s.t. } ||S||\_0 \leq k$, rank$(L\_Q)\leq r, L\_Q$ is quantized
>
> We are currently investigating the three different approaches to determine which would ultimately lead to the best-unified compression approach.

---

> ### Author Response · Authors · 2024-11-17
>
> > By scaling the weight matrix  with a diagonal rescaling matrix , OATS emphasizes outliers, enabling outlier-aware compression that avoids performance degradation.
>
> We thank the reviewer for highlighting a strength with our work and we wanted to mention that we have further elaborated on this strength with a new section Appendix A.3 titled “Using a Robust Scaling Matrix”, included below, that empirically shows that the sensitivity to outliers is needed for good compression:
>
> _To explore whether the scaling matrix $D$ is truly related to the outlier information, we run the following two experiments:_
> 1. _scaling by the square root of the features' second moments, as is currently done in OATS._
> 2. _scaling by the median of the features' absolute values (computed along batch and sequence dimensions):_
>
> $\qquad \qquad \qquad \qquad \qquad D_{robust} =$median$(|X|) $
>
> _The second experiment estimates the square root of the second moment of features in a manner that is robust (insensitive) to outliers akin to the Median Absolute Deviation estimator from the robust statistics literature [1]. The results of the two experiments are presented in the table below:_
>
> | Scaling Matrix   | MMLU (↑) | Zero-shot (↑) | Perplexity (↓) |
> |------------------|----------|---------------|----------------|
> | $D\_{robust}$ | 55.54    | 65.77         | 18.59          |
> | $D $          | **59.99** | **68.41**     | **15.18**      |
>
> _The findings show that using the robust scaling method results in significantly worse performance. Hence, the scaling matrix $D$ that is sensitive to the outlier features and captures their scale leads to better compression._
>
> [1] Huber, P. J. (1981). Robust statistics.
>
> >
>
> We would like to give thanks to the reviewer for their detailed comments, careful inspection of our work, and foresight in extending OATS to quantization. Should the reviewer find it appropriate, the authors would always be appreciative of any additional reconsideration of the score.

---

> ### Author Response · Authors · 2024-11-22
>
> Dear reviewer, we would greatly appreciate it if you could take a moment to read our rebuttal before the deadline as it would give us enough time to construct a detailed response should there be any lingering concerns or issues that could be resolved to improve our work and its score. We are grateful of all the time and energy that the reviewer has dedicated to the demanding nature of the review process and we would like to reiterate our thanks to the reviewer.

---

> > ### Comment · Reviewer_Gg5U · 2024-11-26
> >
> > I appreciate the authors' thorough explanations and efforts to address the questions and concerns raised. I believe OATS presents an interesting contribution to the community, with many promising directions for future research stemming from it. I will maintain my score and advocate for the acceptance of this paper.

---

> ### Author Response · Authors · 2024-12-04
> **Thank You**
>
> Dear Reviewer Gg5U,
>
> As the discussion period comes to a close, we would like to express our sincere gratitude to the reviewer for engaging with us during the discussion period, for the reviewer’s recognition of OATS as an interesting contribution to the community with many promising directions for future research, and for advocating for the acceptance of our work into the conference. Thank you.
>
> Best,
>
> The Authors.

---

### Public Comment · ~徐天宇1 · 2024-11-13
**Question about CPU throughput tests presented in Section 3.4**

I hope this message finds you well.

I have a question regarding the CPU throughput tests presented in Section 3.4 of your paper. Specifically, I was wondering if any hardware acceleration techniques, such as compression or other methods, were used during these tests. If not, in scenarios where the total data volume (SUV matrix > Weight matrix) is larger, what strategies would you suggest for improving throughput?

Thank you for your time, and I look forward to your insights.

Best regards

---

> ### Author Response · Authors · 2024-11-21
>
> Thank you for showing interest in our work. Beyond compressing the weights according to OATS, we did not perform any other compression techniques. However, we did utilize Neural Magic's Deepsparse Engine to benchmark the throughput through the sparse model which utilizes additional techniques to leverage sparsity for speed-up [1]. Two simple solutions to further reduce the memory footprint associated with the unstructured sparse matrix is to either induce more (unstructured) sparsity or to replace unstructured with structured sparsity – both of which OATS is able to do over prior unstructured pruning techniques. However, beyond OATS, another option would be to incorporate quantization which is a direction that we are actively exploring.
>
> [1] https://github.com/neuralmagic/deepsparse

---

### Meta-Review · Area_Chair_APzu · 2024-12-20

**Metareview:**

**Summary**

The paper presents OATS (Outlier-Aware Pruning Through Sparse and Low-Rank Decomposition), a technique developed to compress large transformer models and reduce memory and compute costs without requiring costly retraining. This method is based on decomposing the weight matrices into a combination of a sparse matrix and a low-rank matrix. OATS enhances model compression by scaling the weights of transformer models according to the second moment of input embeddings, a strategy designed to retain crucial outlier features. This method effectively maintains model performance, addressing the common issue of performance degradation observed with higher compression levels in conventional pruning techniques. The authors demonstrate that OATS effectively compresses transformer models for both language and vision tasks, surpassing the performance of other pruning methods across various tasks and compression levels. Additionally, OATS provides notable speed enhancements when deployed on CPUs.

**Strengths**

The reviewers unanimously highlighted several strengths of the proposed framework:
* OATS eliminates the need for retraining, which is essential for practical scenarios where even a single backpropagation pass can be computationally expensive.
* The method has been evaluated on cutting-edge architectures such as Llama and ViT on a large range of model sizes, solidly demonstrating that it can compete with recent benchmarks in terms of performance.
* The paper includes data on practical speed improvements on CPUs when using current frameworks for sparse computation.

**Weaknesses**

The reviewers brought up several core weaknesses:
* Some very relevant prior works are missing, some of which are conceptually identical to the proposed work, e.g., Yu et al., "On compressing deep models by low rank and sparse decomposition," CVPR 2017.
* The choice of rank ratio parameter and why it should be fixed across different layers, as opposed to considering dynamic ranks across different layers, e.g., as done in the concurrent work [1].
* The lack of computational analysis for OATs
* Poor placement of the paper in the landscape of current literature.

[1] "Dynamic Low-Rank Sparse Adaptation for Large Language Models," in Proc. Thirteenth Int. Conf. Learning Representations, 2024, under review. [Online]. Available: https://openreview.net/forum?id=oXh0939Zzq

**Conclusion**

The majority of the reviewers evaluated this paper positively. However, Reviewer BUra (Rating: 3, Confidence: 5) raised several critical issues about the submitted work, with which I largely agree, particularly the omission of critical prior works that are conceptually similar to the current submission. The authors have addressed most of the concerns raised by Reviewer BUra, which unfortunately, Reviewer BUra did not acknowledge. While I share some of Reviewer BUra's reservations, I believe that in light of the authors' rebuttal, the paper's strengths outweigh its weaknesses. Considering this, I view the paper as marginally above the acceptance threshold and recommend acceptance.

**Additional Comments On Reviewer Discussion:**

Despite my efforts to engage the reviewers in a discussion during the review period to reach a consensus on the paper’s merits and shortcomings, there was no participation in any discussion.

Reviewer BUra raised significant concerns about the paper, to which the authors responded adequately. In my view, their responses partially address the issues highlighted by Reviewer BUra. Unfortunately, the reviewer did not acknowledge the authors' rebuttal. While I agree with some of Reviewer BUra's criticisms, considering the authors' rebuttal and aligning with the majority of the reviewers, I believe the strengths of the paper outweigh its shortcomings. Furthermore, given the significance and relevance of the research topic, I assess the paper to be above the acceptance threshold and vote for its acceptance.

---

### Decision · Program_Chairs · 2025-01-22

Accept (Poster)